# Midtraining Bridges Pretraining and Posttraining Distributions

**Emmy Liu** [1]  **Graham Neubig** [1]  **Chenyan Xiong** [1]

## Abstract

Midtraining, the practice of mixing specialized data with more general pretraining data in an intermediate training phase, has become widespread in language model development, yet there is little understanding of what makes it effective. We propose that midtraining functions as distributional bridging by providing better initialization for posttraining. We conduct controlled pretraining experiments, and find that midtraining benefits are largest for domains distant from general pretraining data, such as code and math, and scale with the proximity advantage the midtraining data provides toward the target distribution. In these domains, midtraining consistently outperforms continued pretraining on specialized data alone both in-domain and in terms of mitigating forgetting. We further conduct an investigation on the starting time and mixture weight of midtraining data, using code as a case study, and find that time of introduction and mixture weight interact strongly such that early introduction of specialized data is amenable to high mixture weights, while late introduction requires lower ones. This suggests that late introduction of specialized data outside a plasticity window cannot be compensated for by increasing data mixtures later in training. Beyond midtraining itself, this suggests that distributional transitions between any training phases may benefit from similar bridging strategies. [1]

## 1. Introduction

The success of large language models has mostly been driven by scaling model and data size. Though many interventions seem promising, they may wash out at scale. Therefore, when methodological interventions are simple

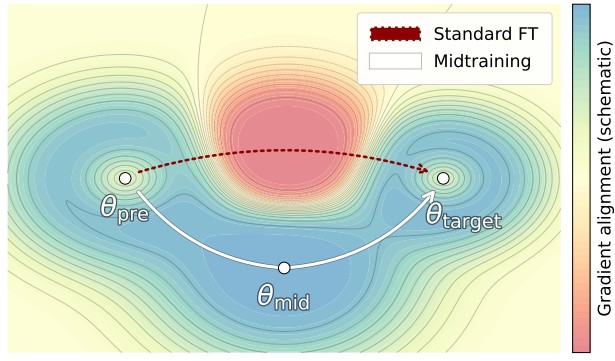

*Figure 1.* Schematic optimization landscape, where contours indicate gradient conflict between pretraining and posttraining objectives. Standard pretraining→SFT follows the red path, while pretraining→midtraining→SFT follows the white path. Midtraining shifts the initialization so SFT can approach the target while avoiding high-conflict regions.

yet widely adopted across model scales, they merit attention. One such intervention is *midtraining*: breaking pretraining into two or more stages in which the latter stages incorporate higher-quality data from specialized domains such as mathematics and coding, as well as instruction-formatted data (Hu et al., 2024; Dubey et al., 2024; OLMo Team et al., 2025). While widely adopted, it is often treated as a heuristic "cooldown" phase, with surprisingly little systematic study of its underlying mechanics or optimal design (Wang et al., 2025).

This raises key questions: is midtraining simply a form of teaching to the test by learning the target distribution, or does it serve a more distinct role in the training trajectory? When is it expected to help performance in-domain, and does it have an impact on forgetting of the pretraining distribution? How does it compare to the similar concept of continued pretraining? To answer these questions, we conduct the first systematic investigation of midtraining, controlling for data domains, mixture ratios, and the timing of this phase relative to overall training schedule.

Our results show that midtraining appears to act as a distributional bridge, smoothing the optimization path from general pretraining data to specialized domains. By systematically varying training conditions, we identify specific conditions under which midtraining is most effective:

---

[1]Language Technologies Institute, Carnegie Mellon University, USA. Correspondence to: Emmy Liu <emmy@cmu.edu>.

*Proceedings of the 43$^{rd}$ International Conference on Machine Learning*, Seoul, South Korea. PMLR 306, 2026. Copyright 2026 by the author(s).

[1]Data and code are available at https://github.com/nightingal3/midtraining.

- **Midtraining yields the largest gains on domains that are "distant" from the general pretraining distribution**, such as mathematics and code. In these high-shift regimes, midtraining appears to mitigate the gradient conflicts that typically hamper standard fine-tuning.

- **Midtraining reduces catastrophic forgetting** compared to both direct fine-tuning and naive continued pretraining. This challenges the "memorization" hypothesis, as the method preserves prior knowledge better than simply training on the target data at the end.

- **The timing of data introduction is often more impactful than the mixture weight itself**. This aligns with a "Plasticity Window" hypothesis, suggesting that midtraining is most effective when applied while the model's representations remain sufficiently malleable to adjust to the new distribution without rigidity.

Taken together, these findings offer a more nuanced view of midtraining: not merely as a final polishing step, but as a geometric intervention that when timed correctly allows models to specialize in distant domains while mitigating forgetting of general knowledge.

## 2. Preliminaries

In this section, we define what we refer to as midtraining throughout this paper. While this term has been used colloquially by model developers, it lacks a standard definition, so we establish our working definition for clarity.

### 2.1. Training Sequence Definitions

Language model training can be viewed as a sequence of phases $S = \{D_i, J_i\}_{i=0}^N$, where parameters $\theta_i$ are initialized from training on data up to phase $i-1$. Standard LM training typically consists of first **pretraining** on a massive, diverse corpus $D_{\text{pre}}$, followed by **posttraining** on a target dataset $D_{\text{target}}$, where often $D_{\text{target}}$ is orders of magnitude smaller and usually on a narrower topic semantically.

We define **midtraining** as any intermediate phase between training stages, in this case between pretraining and posttraining. Midtraining data is typically more specialized than general pretraining data, often including domain-specific content (code, math) and instruction-formatted data, while maintaining a mixture with general pretraining data. Typically, midtraining is a longer phase compared to fine-tuning, but shorter than the preceding pretraining phase, $|D_{\text{pre}}| > |D_{\text{mid}}| > |D_{\text{target}}|$. There can potentially be multiple midtraining phases as well in the case of multi-stage pretraining curricula, but we focus on one-stage midtraining in this paper.

### 2.2. Relationship to Curriculum Learning and Continued Pretraining

**Curriculum learning** The original definition of curriculum learning focused on gradually increasing the difficulty or diversity of training examples throughout the course of training (Elman, 1993; Bengio et al., 2009). However, this term has evolved to generally mean any strategic ordering of training data (Soviany et al., 2022). Midtraining can be viewed as a coarse-grained distributional curriculum; instead of ordering individual examples, it orders data distributions across discrete training phases.

**Continued pretraining** Continued pretraining adapts a pretrained model by training further on domain-specific data, typically with a full shift to the target distribution (Gururangan et al., 2020; Beltagy et al., 2019). This can improve in-domain performance but risks degrading general capabilities. Midtraining differs in that it retains a mixture with general pretraining data during the intermediate phase. In our setup, continued pretraining is the limiting case where the mixture weight on general pretraining data is zero. In practice, both are often implemented as additional next-token-prediction training, differing mainly in the degree of distribution shift and associated schedule/optimizer choices.

### 2.3. Theoretical Analysis

In the previous section, we defined midtraining as an intermediate phase that mixes general pretraining data with a specialized distribution before posttraining. Here, we give a simple theoretical sketch that formalizes the core intuition behind our empirical results: midtraining acts primarily through the initialization for posttraining, and this can simultaneously (i) improve in-domain posttraining loss and (ii) mitigate forgetting on the pretraining distribution. Full derivations appear in Appendix A.

**Setup.** Let $J_P(\theta)$ denote the population loss on the pretraining distribution $P$, and $J_T(\theta)$ the loss on the posttraining/SFT distribution $T$. Posttraining runs $K$ steps of gradient descent on $J_T$ starting from an initialization $\theta_0$:

$$\theta_{k+1} = \theta_k - \eta \nabla J_T(\theta_k), \qquad k = 0, \ldots, K-1, \quad (1)$$

and we measure posttraining-induced forgetting by the increase in pretraining loss, $\Delta_P(K) := J_P(\theta_K) - J_P(\theta_0)$. Here $\theta_0$ is the checkpoint at which posttraining begins; when it is obtained by midtraining, we write $\theta_0 = \theta_0(t, w)$ to emphasize its dependence on the midtraining start time $t$ and mixture weight $w$. Thus, $\Delta_P(K)$ excludes any change in pretraining loss incurred during midtraining itself, which we measure separately in our experiments.

**Forgetting decomposition (smoothness sketch).** Assume $J_P$ is $L_P$-smooth. Consider $K$ steps of posttrain-

ing GD on $J_T$, $\theta_{t+1} = \theta_t - \eta \nabla J_T(\theta_t)$. Applying the standard smoothness upper bound to $J_P$ at $\theta_t$ with step $\theta_{t+1} - \theta_t = -\eta \nabla J_T(\theta_t)$ gives the one-step inequality

$$
\begin{aligned}
J_P(\theta_{t+1}) - J_P(\theta_t) \leq & -\eta \langle \nabla J_P(\theta_t), \nabla J_T(\theta_t) \rangle \\
& + \frac{L_P \eta^2}{2} \| \nabla J_T(\theta_t) \|^2.
\end{aligned}
\tag{2}
$$

Summing (2) over $t = 0, \ldots, K-1$ yields a telescoping sum:

$$
\Delta_P(K) := J_P(\theta_K) - J_P(\theta_0),
$$

$$
\Delta_P(K) \leq \underbrace{-\eta \sum_{t=0}^{K-1} \langle \nabla J_P(\theta_t), \nabla J_T(\theta_t) \rangle}_{\text{(A) alignment / conflict along the posttraining path}}
$$

$$
+ \underbrace{\frac{L_P \eta^2}{2} \sum_{t=0}^{K-1} \| \nabla J_T(\theta_t) \|^2}_{\text{(B) posttraining ``energy'' (squared-gradient) term}}.
\tag{3}
$$

**Relating (B) to posttraining progress.** If $J_T$ is $L_T$-smooth and $\eta \leq 1/L_T$, a standard GD descent inequality gives $J_T(\theta_{t+1}) \leq J_T(\theta_t) - \frac{\eta}{2} \| \nabla J_T(\theta_t) \|^2$. Summing over $t$ yields

$$
\begin{aligned}
\sum_{t=0}^{K-1} \| \nabla J_T(\theta_t) \|^2 & \leq \frac{2}{\eta} \big( J_T(\theta_0) - J_T(\theta_K) \big) \\
& \leq \frac{2}{\eta} \big( J_T(\theta_0) - J_T^\star \big),
\end{aligned}
\tag{4}
$$

where $J_T^\star := \inf_\theta J_T(\theta)$.

Finally, we can substitute back to get the forgetting bound:

$$
\Delta_P(K) \leq \underbrace{-\eta \sum_{t=0}^{K-1} \langle \nabla J_P(\theta_t), \nabla J_T(\theta_t) \rangle}_{\text{(A) alignment term}}
$$

$$
+ \underbrace{L_P \eta \big( J_T(\theta_0) - J_T^\star \big)}_{\text{(B) effort term}}.
\tag{5}
$$

**Connection to midtraining.** Because midtraining changes only the initialization $\theta_0 = \theta_0(t, w)$, it can change the upper bound on forgetting by providing an initialization which has to do less work initially ($J_T(\theta_0) - J_T^\star$ smaller).

## 3. Experimental Setting

Having defined midtraining as an intermediate phase between pretraining and posttraining, we next specify the controlled experiments we use to study this training phase. Across our experiments, we keep the model family and posttraining procedure fixed and vary the conditions of midtraining. We organize this section around four key research questions, which we introduce one by one along results.

| Midtrain mix | Num. Tokens (B) | Sources |
|---|---|---|
| Starcoder | 196 | (Li et al., 2023) |
| Math | 12 | (Yue et al., 2023; Toshniwal et al., 2024) |
| FLAN | 3.5 | (Wei et al., 2022) |
| KnowledgeQA | 9.6 | (Hu et al., 2024) |
| DCLM | 51 | (Li et al., 2024b) |

*Table 1.* Midtraining mixes used in our experiments and dataset(s) from which they were derived.

### 3.1. Training Setup

**Pretraining** We pretrain models from the Pythia family ranging in size from 70M-1B parameters on C4 web data (Raffel et al., 2020; Biderman et al., 2023). In all cases, we train for 128B tokens (approx. 61k steps) with a cosine learning rate schedule with a maximum learning rate of 3e-4 and the AdamW optimizer (Loshchilov & Hutter, 2019). We chose to fix the training budget at a point past Chinchilla-optimality for all models (Hoffmann et al., 2022), in order to ensure that models have stabilized by the point at which midtraining data has been introduced, at least for later insertion points of midtraining data. We describe the exact training setup in Appendix B.

**Midtraining** We use five midtraining mixtures spanning popular domains: code (Starcoder), math, instructions (FLAN), general knowledge/QA, and high-quality web data (DCLM). Table 1 details each mixture's composition and sources. All mixtures are introduced at varying start points (Starcoder: 6k steps, Math: 20k steps, others: 40k steps) based on data availability to prevent repetition. We compare against a control condition continuing C4 pretraining for the same number of tokens, keeping all other training details identical.

**Starcoder (code)** Our code mixture is a subset of the Starcoder pretraining dataset (Li et al., 2023), which contains code in many languages. Note that we use code from all languages, rather than Python.

**Math** The math mixture combines mathematical reasoning problems from the MAmmoTH (Yue et al., 2023) and OpenMathInstruct (Toshniwal et al., 2024) datasets, featuring step-by-step explanations.

**FLAN (instructions)** Our instruction-formatted data comes from a processed version of the FLAN collection, which includes diverse task instructions and responses across natural language tasks (Wei et al., 2022).

**KnowledgeQA (general knowledge and QA)** The KnowledgeQA mixture is taken from Hu et al. (2024)'s midtraining mix, and focuses on general knowledge and dialogue. However, to distinguish the midtraining mixes fur-

ther, the StackOverflow portion of this dataset is removed.

**DCLM (high-quality web)**   Our high-quality web data is a subset of the DCLM pretraining dataset, representing web content with improved quality filtering compared to C4 (Li et al., 2024b).

**Downstream Evaluation**   We fine-tune models on the datasets GSM8k (Cobbe et al., 2021), SciQ (Welbl et al., 2017), CodeSearchNet-Python (Husain et al., 2019), and LIMA (Zhou et al., 2023) – chosen to span the domains covered by our midtraining mixtures. This allows us to test cases where the midtraining mixture is aligned or misaligned with the SFT dataset. We used standard language model supervised fine-tuning for all datasets. For information on the posttraining setup, see Appendix C.

**Catastrophic Forgetting Evaluation**   A key concern with supervised fine-tuning is whether introducing specialized data causes models to forget general capabilities acquired during pretraining. We measure catastrophic forgetting by evaluating cross-entropy loss on the original pretraining distribution by measuring loss on held-out C4 data. This approach follows established practices for measuring forgetting in language models (Luo et al., 2024; Kemker et al., 2018; Li et al., 2024a).

**Proximity advantage.**   To quantify whether a midtraining mixture moves the training distribution toward a target SFT dataset, we compute a token-level proximity score $\mathrm{prox}(\cdot, \cdot)$ between corpora using unigram token statistics under the model tokenizer (example in Figure 2, full in Appendix D). Given a target dataset $T$ and a midtraining mixture $M$, we define the *proximity advantage* of $M$ relative to continuing pretraining on C4 as

$$\mathrm{PA}(M \to T) = \mathrm{prox}(M, T) - \mathrm{prox}(\mathrm{C4}, T). \quad (6)$$

Positive PA indicates that $M$ is closer to $T$ than C4 at the token level. While our theory is stated in terms of optimization quantities (e.g., gradient alignment), PA provides an inexpensive, model-agnostic diagnostic of distribution shift.

## 4. Which downstream tasks benefit most from midtraining?

We begin by asking **where** midtraining is most effective. We evaluate all combinations of midtraining mixtures and SFT targets, reporting (i) target-domain validation loss after SFT (adaptation) and (ii) C4 validation loss after SFT (forgetting). We average results over 5 seeds after hyperparameter search for each checkpoint.

We find that across model sizes, midtraining benefits are highly domain-specific: specialization on code yields the largest gains on code tasks, while math-focused midtraining helps mathematical-reasoning tasks. Mismatched midtraining provides minimal benefit, and general instruction mixes (e.g., FLAN) produce little improvement. Full per-dataset results and numerical comparisons are reported in Table 2 and Appendix E. Interestingly, in this setting, in-domain improvements and C4 retention are strongly aligned (Pearson $r = 0.64, p \approx 3.7 \times 10^{-12}$).

To separate the effect of midtraining on the initialization from forgetting caused by SFT, we additionally measure losses before SFT in the main matched settings. As shown in Table 3, code and math midtraining substantially improve the corresponding target-domain loss before SFT while leaving held-out C4 loss nearly unchanged. Using these pre-SFT measurements, we compute posttraining-only forgetting as the increase in C4 loss from before to after SFT. Matched midtraining reduces this SFT-induced C4 degradation from 0.073 to 0.038 for Pycode, and from 0.132 to 0.020 for GSM8K. Thus, matched midtraining provides a better target-domain initialization while also reducing the forgetting incurred during the subsequent SFT stage.

> **Finding 1.**  Midtraining benefits are highly domain-specific. Code-focused midtraining (Starcoder) yields large gains on coding tasks, and math-focused midtraining improves mathematical reasoning. Mismatched domains provide little benefit, and broad instruction data (FLAN) also shows minimal effect.

## 5. What data is most effective for midtraining?

Having established that midtraining effects are domain-specific, we now ask: *what determines the strength of these domain-specific effects*? Across midtraining-target pairs, we see improvements ranging from negligible (e.g. FLAN $\to$ coding) to strong (e.g. Starcoder $\to$ coding). We consider two candidate explanations: (i) whether the midtraining distribution is "closer" to the target than C4 (distributional bridging), and (ii) whether retaining some general data is necessary compared to switching fully to specialized data.

### 5.1. Proximity and Bridging Effects

To understand why some midtraining mixtures are effective, we test the hypothesis that good midtraining data *bridges* the distributional gap between pretraining (C4) and the target SFT dataset. Concretely, we use the proximity advantage $\mathrm{PA}(M \to T)$ defined in Equation 6, the increase in token-level proximity to the target achieved by midtraining on mixture $M$ relative to continuing on C4 (Appendix D). Results are shown in Figure 3.

We find a positive relationship between proximity advantage and downstream performance across model sizes, suggest-

*Table 2.* SFT and C4 validation losses for the 1B model across downstream datasets and midtraining mixtures (5 seeds per SFT dataset). Bold indicates best within each dataset. Percentages denote the specialized data proportion mixed with C4 during midtraining. Parentheses show Δ relative to the C4-only baseline within each dataset (negative is better).

| Dataset | Midtrain Mix | Validation Loss | |
| --- | --- | --- | --- |
| | | SFT | C4 |
| **Pycode** | | | |
| | **C4** | 2.174 (+0.000) | 3.075 (+0.000) |
| | Starcoder (20%) | **1.888** (-0.286) | 3.070 (-0.005) |
| | Math (12%) | 2.175 (+0.000) | **3.057** (-0.018) |
| | FLAN (5%) | 2.174 (+0.000) | 3.083 (+0.008) |
| | KnowledgeQA (20%) | 2.174 (+0.000) | 3.104 (+0.029) |
| | DCLM (20%) | 2.175 (+0.001) | 3.106 (+0.031) |
| **GSM8K** | | | |
| | **C4** | 0.942 (+0.000) | 3.134 (+0.000) |
| | Starcoder (20%) | 0.927 (-0.015) | 3.158 (+0.024) |
| | Math (12%) | **0.851** (-0.091) | **3.018** (-0.116) |
| | FLAN (5%) | 0.942 (+0.000) | 3.135 (+0.001) |
| | KnowledgeQA (20%) | 0.943 (+0.001) | 3.136 (+0.002) |
| | DCLM (20%) | 0.943 (+0.001) | 3.137 (+0.003) |
| **LIMA** | | | |
| | **C4** | 3.316 (+0.000) | 2.900 (+0.000) |
| | Starcoder (20%) | 3.315 (-0.001) | 2.930 (+0.030) |
| | Math (12%) | **3.310** (-0.006) | **2.893** (-0.008) |
| | FLAN (5%) | 3.316 (-0.001) | 2.900 (+0.000) |
| | KnowledgeQA (20%) | 3.315 (-0.002) | 2.900 (+0.000) |
| | DCLM (20%) | 3.314 (-0.002) | 2.900 (-0.001) |
| **SciQ** | | | |
| | **C4** | 1.858 (+0.000) | 3.201 (+0.000) |
| | Starcoder (20%) | 1.873 (+0.015) | 3.245 (+0.044) |
| | Math (12%) | 1.876 (+0.018) | **3.182** (-0.019) |
| | FLAN (5%) | **1.856** (-0.002) | 3.192 (-0.009) |
| | KnowledgeQA (20%) | **1.856** (-0.002) | 3.196 (-0.005) |
| | DCLM (20%) | 1.857 (-0.001) | 3.201 (+0.000) |

*Table 3.* Matched midtraining improves target-domain initialization while largely preserving C4 before SFT, and reduces posttraining-only forgetting.

| Dataset | Quantity | C4 Init. | Matched Init. |
| --- | --- | --- | --- |
| **Pycode** | (matched init.: Starcoder 20%) | | |
| | Pre-SFT target | 4.151 (+0.000) | **1.948** (-2.203) |
| | Pre-SFT C4 | **3.002** (+0.000) | 3.032 (+0.030) |
| | Post-SFT C4 | 3.075 (+0.000) | **3.070** (-0.005) |
| | Post. ΔC4 | +0.073 | +0.038 |
| **GSM8K** | (matched init.: Math 12%) | | |
| | Pre-SFT target | 2.886 (+0.000) | **1.301** (-1.585) |
| | Pre-SFT C4 | 3.002 (+0.000) | **2.998** (-0.004) |
| | Post-SFT C4 | 3.134 (+0.000) | **3.018** (-0.116) |
| | Post. ΔC4 | +0.132 | +0.020 |

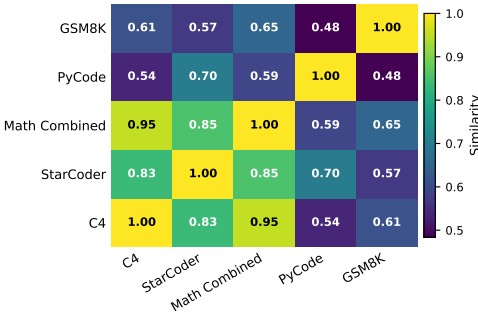

*Figure 2.* Example similarity matrix between pre/midtrain and posttraining datasets. For the complete matrix, see Appendix D.

ing that effective midtraining data serves as a distributional stepping stone from general pretraining to specialized target domains. This bridging effect appears to be most beneficial when the gap between pretraining and target distributions is large, consistent with our hypothesis that midtraining helps models adapt gradually rather than requiring abrupt distributional shifts during fine-tuning.

> **Finding 2.** Midtraining gains are well-predicted by a simple proximity advantage metric: mixtures that are closer to the target than C4 (in terms of token distributions) yield larger improvements.

### 5.2. Midtraining vs. Continued Pretraining

Our results so far suggest that effective midtraining data serves as a bridge between general pretraining and specialized posttraining data. However, a question that follows is why midtraining is necessary: continued pretraining on domain-specific data also aims to adapt the model toward a target domain. Why not simply pretrain normally and then switch to domain-specific data entirely?

To examine this, we compare the effect of midtraining with continued pretraining in which the mixture weight switches to 100% specialized data. For code, we compare the default

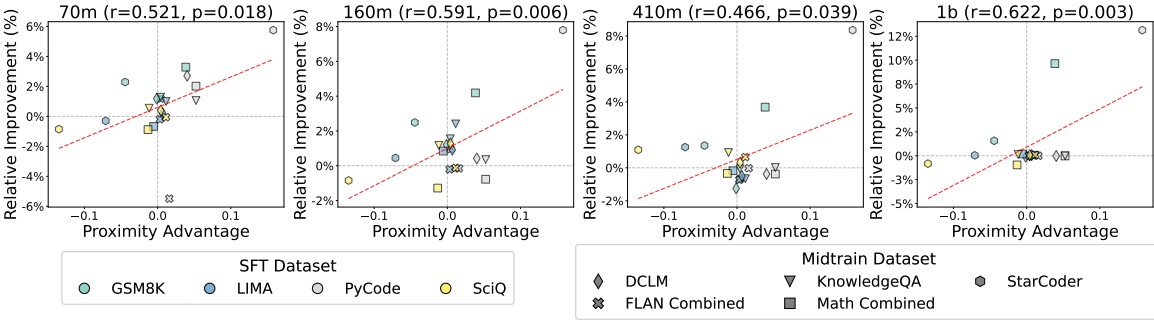

*Figure 3.* Relationship between proximity advantage and midtraining performance improvements for pairs of midtraining and SFT datasets. Each data point represents a (midtrain, SFT) pair, where the color indicates the SFT dataset and shape represents midtrain dataset. Proximity advantage (dist(C4, SFT) - dist(midtrain, SFT)) indicates how much closer midtraining data brings the model to the target SFT dataset compared to the base pretraining data. Proximity advantage pairs near zero are greyed out for clarity but included in calculations. Relative improvement is measured against the base model pretrained on C4.

Starcoder midtraining mix (20% mixture weight, starting from 12.6B tokens) with 100% Starcoder data starting from 83B tokens. For math, we compare the math midtraining mix with 100% math data starting from 105B tokens.[2] This comparison intentionally gives continued pretraining more domain-specific tokens than midtraining. Thus, the result is not driven by midtraining seeing more specialized data.

Results in Table 4 show that midtraining consistently outperforms continued pretraining across both domains and model sizes for both in-domain performance and C4 retention after fine-tuning. As this pattern holds for both code and math domains, this suggests that maintaining some general pretraining data is useful during domain adaptation, even for models specialized for a specific domain. This supports our intuition gained from prior sections that domain adaptation benefits from gradual distributional shifts at the token level rather than abrupt changes.

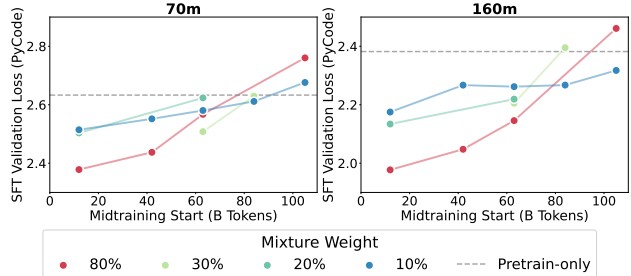

*Figure 4.* Effect of mixture weight and midtraining phase start on in-domain validation loss for the code mixture. A high mixture weight is beneficial when the midtraining phase begins early, but is detrimental when beginning this phase later.

> **Finding 3.** Maintaining a mixture with general data in midtraining is preferable to continued pretraining on specialized data alone.

# 6. When and how much midtraining data should be introduced?

Having established that effective midtraining data bridges syntactic patterns in pretraining and posttraining datasets, we now ask a natural question: when should this bridge be introduced, and how much specialized data should be mixed in? Although practitioners routinely tune midtraining mixture weights, the choice of when to begin midtraining, and critically, how start time interacts with mixture weight—has received little systematic study.

We conduct targeted experiments varying both the start point of the midtraining phase (between 12B and 105B tokens into pretraining) and mixture weight (between 10-80% specialized data). We test multiple combinations of starting point and mixture weight to test hypotheses about the interactions between timing and mixture weight, namely: (1) Do

*Table 4.* SFT and C4 validation losses for 70M and 160M models comparing default midtraining mixes to continued pretraining on only the midtraining data (100%), averaged across 5 seeds. Bold indicates best performance within each dataset/model combination.

| Model | Dataset | Mix | SFT | C4 |
|---|---|---|---|---|
| **70M** | | | | |
| | Pycode | Pretrain-only | 2.656 | 6.152 |
| | | Starcoder (20%) | **2.504** | **6.032** |
| | | Ctd. pretrain (Starcoder) | 2.530 | 6.109 |
| | GSM8K | Pretrain-only | 1.384 | **6.353** |
| | | Math (12%) | **1.339** | 6.358 |
| | | Ctd. pretrain (Math) | 1.383 | 6.376 |
| **160M** | | | | |
| | Pycode | Pretrain-only | 2.314 | 5.254 |
| | | Starcoder (20%) | **2.134** | **5.079** |
| | | Ctd. pretrain (Starcoder) | 2.219 | 5.369 |
| | GSM8K | Pretrain-only | 1.163 | 5.308 |
| | | Math (12%) | **1.114** | **5.230** |
| | | Ctd. pretrain (Math) | 1.159 | 5.326 |

---

[2]The different starting points are due to data availability, to ensure the midtraining mix does not repeat.

timing and mixture weight interact, or do they have independent effects? (2) Can later introduction of specialized data be compensated for by increasing mixture weight? We conduct experiments on the 70m and 160m models with the Starcoder mixture to study these questions, as it is the mix with the strongest in-domain effects.

**Timing and mixture weight interact strongly.** Figure 4 shows that the optimal mixture weight of specialized data depends critically on when the midtraining phase begins. Early introduction of code with a very high mixture weight (80%) achieves the best in-domain performance. However this relationship reverses later in training, with the high mixture weight (80%) performing substantially worse than the conservative mixture (10%) at 105B tokens.

**Compensation through increased mixture fails.** Suppose that we have already pretrained a model to a certain number of tokens, and do not want to redo training to accommodate a new midtraining component. Can we make up for this through using higher mixture weights at later start times? When examining the progression of loss values (10% @ 42B, 20% @ 63B, 30% @ 84B), we can see that this is not the case, as shifting from an early introduction point and conservative mixture weight to a late introduction point and aggressive mixture weight degrades performance. This suggests that the model may lack sufficient plasticity to adapt to a high weight of specialized data late in pretraining. However, we note that this is a small-scale ablation: while the trends are consistent for smaller models, larger-scale sweeps are needed to fully validate this finding.

Relatedly, Figure 5 illustrates how midtraining benefits evolve over the course of pretraining for the 20% Starcoder mix (160m model). We finetune checkpoints from different starting points on Pycode and measure both in-domain and C4 validation loss after fine-tuning. In-domain advantages emerge quickly after midtraining introduction (12.6B tokens), while the C4 retention benefits develop more gradually, becoming apparent after approximately 42B tokens. This temporal pattern suggests that early introduction of specialized data provides sufficient time for both immediate domain adaptation and gradual integration with general capabilities.

> **Finding 4.** Midtraining effectiveness depends on a *timing-weight interaction*. For code midtraining, early introduction supports aggressive mixture weights and yields the largest in-domain gains, whereas late introduction makes high mixture weights harmful and favors more conservative mixing.

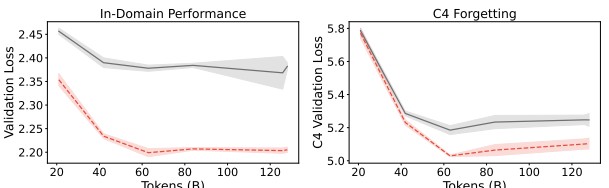

*Figure 5.* Validation loss and C4 loss for the Starcoder-midtrained model (160M), (red line) and base pretrained model (grey line) *after* supervised fine-tuning on the Pycode dataset, with each point on the x-axis representing the number of tokens the pretrained checkpoint was trained on.

## 7. How does midtraining change model representations?

Our results suggest that midtraining can improve both in-domain adaptation and pretraining retention when the midtraining mix is well-aligned to the target dataset. We next ask whether this is reflected in the representational changes models undergo during fine-tuning. As an initial descriptive probe, we compare representations between midtrained and base models in the code domain.

We use linear Centered Kernel Alignment (CKA) to measure layer-wise similarity between model states (Kornblith et al., 2019). We extract activations from all layers using probe datasets (C4 and APPS (Hendrycks et al., 2021)) and compute CKA similarity matrices between four key model states: base pretrained, midtrained (Starcoder), base fine-tuned, and midtrained fine-tuned. If midtraining provides a better initialization for downstream tasks, we expect to see smaller representational changes during fine-tuning for midtrained models compared to base models.

Figure 6 shows the representational analysis for the 70M model. The midtrained model changes less in the final layer after fine-tuning, a pattern consistent across model sizes (see Appendix G for the remaining results). However, the final fine-tuned models show high similarity regardless of whether models underwent midtraining. Therefore, we do not interpret CKA as evidence that midtraining leads to a fundamentally different final representation, but rather a different fine-tuning trajectory. These effects are less pronounced for C4, which can be seen in Appendix H. Overall, this is qualitatively consistent with the view that midtraining acts as a better initialization for downstream SFT, reducing the amount of late-layer change needed to reach a similar fine-tuned representation.

> **Finding 5.** Models exposed to midtraining require smaller representational shifts during fine-tuning, especially in the final layer, indicating smoother adaptation.

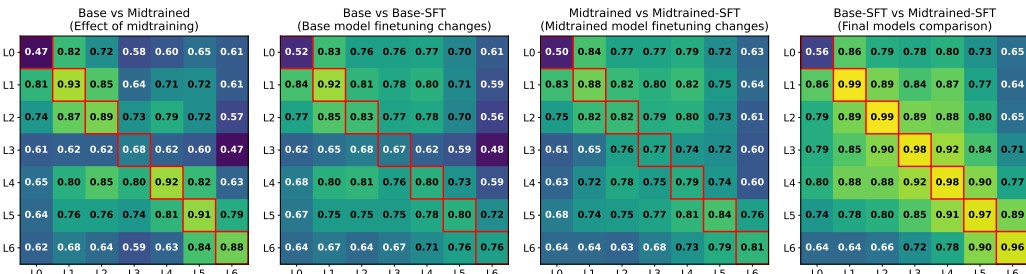

*Figure 6.* CKA analysis of model activations in the 70M model, probed with the APPS code dataset.

# 8. Related Work

**Specific midtrained models**   Recently, several language model families have adopted midtraining approaches with varying implementation details (Hu et al., 2024; Dubey et al., 2024; OLMo Team et al., 2025; Olmo et al., 2026; Chameleon Team, 2024). The midtraining phase duration varies from 2% (Hu et al., 2024) to 20% (Chameleon Team, 2024) of total training, motivating our systematic investigation of timing effects. Common midtraining domains include code, math, instructions, and higher-quality web data (OLMo Team et al., 2025)—the domains we investigate. Beyond general-purpose models, midtraining has shown benefits for specific tasks like RL (Wang et al., 2025) and GUI agents (Zhang et al., 2025a). This widespread adoption motivates our questions of when and why midtraining provides downstream benefits.

**Staged training and pre-adaptation**   Several works explore multi-stage pretraining, Feng et al. (2024); Blakeney et al. (2024) focusing on two-stage pretraining and Zhang et al. (2025b) proposing four-stage pretraining. These approaches demonstrate improvements over single-stage pretraining. However, these works evaluate base model performance after pretraining, whereas we focus on the post-finetuning setting to focus on benefits that also affect posttraining. Relatedly, domain-adaptive pretraining (DAPT) and related approaches continue pretraining on domain-specific data (Gururangan et al., 2020). Krishna et al. (2023) show that pretraining on downstream data alone can rival full pretraining when evaluated after fine-tuning, suggesting pretraining-posttraining alignment matters—consistent with our findings. Mehta et al. (2023) find pretraining reduces catastrophic forgetting during sequential fine-tuning; similarly, we observe midtrained models serve as better initializations with less forgetting. Most similar in spirit to our bridging interpretation is work on pre-finetuning, which selects unlabeled intermediate data to shift a model's training distribution toward the downstream target (Kang et al., 2024). However, compared to selection-focused approaches, we treat midtraining as a training phase in its own right. Our results show that these schedule choices impact downstream gains and retention, and are based on a similar principle to data selection work.

**Concurrent mixing in posttraining**   A complementary line of work replays pretraining-distribution data during finetuning in order to regularize training, with the goal of mitigating catastrophic forgetting. These include methods that inject selected pretraining samples during finetuning (Liu et al., 2022) as well as rehearsal schemes for multi-stage finetuning (Bai et al., 2025). Scaling analyses have also characterized forgetting as a predictable function of model scale, target data size, and percentage of replay data (Bethune et al., 2025). Although this concept may be similar, replay during fine-tuning intervenes at a different point in the training trajectory: it looks backward, mixing pretraining data as a stability constraint while optimizing the terminal adaptation objective. Midtraining instead looks forward, shaping the initialization so subsequent posttraining is both more effective and less destructive. The two approaches may also be used together.

**Stability and Plasticity in training dynamics**   Recent work addresses stability challenges during continued pretraining. Guo et al. (2024) identify a "stability gap" where performance temporarily drops before recovering when shifting to new domains, Yang et al. (2024) synthesize larger training corpora from small domain-specific datasets, and Lin et al. (2024) introduce selective training on useful tokens only. While these works target training dynamics during continued pretraining, our approach examines how midtraining data selection affects post-fine-tuning performance, representing a complementary focus on end-task effectiveness.

**Relationship between Pretraining and Finetuning**   Several recent works have explored incorporating instruction-formatted data during pretraining. Allen-Zhu & Li (2023) show with an experiment on synthetic Wikipedia-style data that augmenting pretraining data with QA-formatted data improves subsequent fine-tuning, and Jiang et al. (2024) and Cheng et al. (2024) demonstrate this in a practical context as well. Sun & Dredze (2024) find continual pretraining

benefits emerge only after fine-tuning, while Springer et al. (2025) show extended pretraining causes catastrophic forgetting ("overtraining"), particularly on math/code domains least aligned with web data. It is possible midtraining may prevent overtraining by introducing specialized data earlier and providing a better initialization for posttraining.

## 9. Conclusion

We conduct a systematic investigation of midtraining through controlled experiments. We demonstrate that midtraining benefits are domain-specific, with the most substantial improvements in math and code domains that are not well represented in standard web pretraining corpora. Furthermore, we also find that midtraining mitigates catastrophic forgetting of general language modeling abilities after specific supervised fine-tuning and consistently outperformed continued pretraining on specialized data alone. Additionally, timing and mixture weight interact, such that the effectiveness of higher mixture weights depends on when specialized data is introduced.

Practically, these results suggest targeting midtraining toward domains whose token patterns differ substantially from base pretraining data, especially when those domains will be used for posttraining. They also motivate exploring timing of data introduction more systematically, and favoring midtraining over continued pretraining. Looking ahead, it will be important to test whether these trends persist at larger scales and across a broader range of domains, and to understand extensions to reinforcement-learning-based posttraining and multi-stage curricula.

## Impact Statement

This work studies midtraining as a mechanism for improving the effectiveness of sequential training on different data distributions. We anticipate that by clarifying when and how specialized data should be introduced, our findings may reduce trial-and-error experimentation and thus lower the compute costs required to train and adapt language models to new domains. Our experiments use established datasets and do not introduce new data collection or deployments, however as always, practitioners should continue to follow best practices for data governance and copyright when selecting midtraining data.

## Acknowledgements

We thank Scott Wen-tau Yih for mentorship on an earlier version of this project. We also thank Cathy Jiao, Xiaochuan Li, Shanshan Zhong, and Zichun Yu for helpful feedback and advice on this paper. EL was supported by the National Sciences and Engineering Research Council of Canada (NSERC), [funding reference number 578085], as well as the SoftBank-ARM Fellowship.

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

# A. Theoretical Analysis

Here, we present a simple theoretical analysis on the influence of midtraining on forgetting of the original data distribution, as well as on in-domain loss. We use a minimal set of assumptions commonly used in first-order optimization analyses in order to obtain a tractable bound. However, we do not claim that these assumptions hold globally for large language models.

Let the population loss on the pretraining distribution be $J_P(\theta)$ and the population loss on the SFT distribution $T$ be $J_T(\theta)$. Let $\theta_0$ be the parameters of a model immediately before finetuning, and let $\theta_K$ represent the SFT loss after K steps of gradient descent on the SFT dataset.

## A.1. Assumptions and Notation

**Assumption A.1** (Local smoothness). *$J_P$ is $L_P$-smooth and $J_T$ is $L_T$-smooth on a neighborhood containing the iterates $\{\theta_k\}_{k=0}^K$, and on line segments between them i.e., $\|\nabla J(\theta) - \nabla J(\theta')\| \le L\|\theta - \theta'\|$ for $J \in \{J_P, J_T\}$.*

**Assumption A.2** (Step size). *The posttraining step size satisfies $\eta \le 1/L_T$.*

## A.2. Standard inequalities

We use two standard consequences of $L$-smoothness:

**Lemma A.3** (Quadratic upper bound / descent lemma). *If $f$ is $L$-smooth on a convex domain, then for all x, y in the domain,*

$$f(y) \le f(x) + \langle \nabla f(x), y - x \rangle + \frac{L}{2}\|y - x\|^2. \tag{7}$$

**Lemma A.4** (GD decrease lemma). *If $f$ is $L$-smooth and $\eta \le 1/L$, then for the GD update $\theta^+ = \theta - \eta\nabla f(\theta)$,*

$$f(\theta^+) \le f(\theta) - \frac{\eta}{2}\|\nabla f(\theta)\|^2. \tag{8}$$

Assume that the model undergoes a midtraining stage at some point before finetuning, where $\theta_0(t, w)$ represents the model after midtraining, with start timestep $t$ and mixture weight $w$. Additionally, define $\theta_t^{\text{pre}}$ as the parameters immediately before midtraining, and the let $\delta(t, w) = \theta_0(t, w) - \theta_t^{\text{pre}}$ be the change in parameters during the midtraining phase.

We are interested in the **forgetting** of the model after $K$ steps of finetuning, namely:

$$\Delta_P(K) := J_P(\theta_K) - J_P(\theta_0) \tag{9}$$

## A.3. Bounding forgetting over $K$ steps

We start by bounding a one-step change in $J_P$ in the direction $\theta_{t+1} = \theta_t - \eta\nabla J_T(\theta_t)$. Because $J_P$ is $L_P$ smooth, we can apply the usual Equation 7:

$$J_P(\theta_{t+1}) \le J_P(\theta_t) - \eta\langle\nabla J_P(\theta_t), \nabla J_T(\theta_t)\rangle + \frac{L_P\eta^2}{2}\|\nabla J_T(\theta_t)\|^2$$

$$J_P(\theta_{t+1}) - J_P(\theta_t) \le -\eta\langle\nabla J_P(\theta_t), \nabla J_T(\theta_t)\rangle + \frac{L_P\eta^2}{2}\|\nabla J_T(\theta_t)\|^2. \tag{10}$$

We can sum the one-step inequality over $t = 0, \ldots, K - 1$ to yield a telescoping sum:

$$\sum_{t=0}^{K-1}\left(J_P(\theta_{t+1}) - J_P(\theta_t)\right) \le -\eta\sum_{t=0}^{K-1}\langle\nabla J_P(\theta_t), \nabla J_T(\theta_t)\rangle + \frac{L_P\eta^2}{2}\sum_{t=0}^{K-1}\|\nabla J_T(\theta_t)\|^2$$

$$\Delta_P(K) = J_P(\theta_K) - J_P(\theta_0) \le \underbrace{-\eta\sum_{t=0}^{K-1}\langle\nabla J_P(\theta_t), \nabla J_T(\theta_t)\rangle}_{\text{gradient alignment}} + \underbrace{\frac{L_P\eta^2}{2}\sum_{t=0}^{K-1}\|\nabla J_T(\theta_t)\|^2}_{\text{energy term}}. \tag{11}$$

We can see that there is one "gradient alignment" based term and one "energy" based term. We can also bound the "energy" term with Equation 8.

$$J_t(\theta_{t+1}) \leq J_T(\theta_t) - \frac{\eta}{2}\|\nabla J_t(\theta_t)\|^2$$

$$\eta\|\nabla J_t(\theta_t)\|^2 \leq 2(J_t(\theta_t) - J_t(\theta_{t+1}))$$

Again using a telescoping sum, Summing over $t = 0, \ldots, K-1$ yields

$$\eta \sum_{t=0}^{K-1} \|\nabla J_T(\theta_t)\|^2 \leq 2\big(J_T(\theta_0) - J_T(\theta_K)\big)$$

Let $J_T^*$ represent the best possible loss on $T$. Then we can write

$$\eta \sum_{t=0}^{K-1} \|\nabla J_T(\theta_t)\|^2 \leq 2\big(J_T(\theta_0) - J_T^*\big)$$

$$\eta^2 \sum_{t=0}^{K-1} \|\nabla J_T(\theta_t)\|^2 \leq 2\eta\big(J_T(\theta_0) - J_T^*\big)$$

Substituting back into Equation 11, we get the final bound:

$$\boxed{\Delta_P(K) \leq -\eta \sum_{t=0}^{K-1} \langle \nabla J_P(\theta_t), \nabla J_T(\theta_t)\rangle + L_P\,\eta\big(J_T(\theta_0) - J_T^*\big)} \tag{12}$$

We now show how midtraining can potentially impact forgetting through initialization.

**Lemma A.5** (Initialization effect on the energy term). *Assume $J_T$ is $L_T$-smooth on a convex neighborhood containing $\theta$ and $\theta + \delta$. Then for any displacement $\delta$,*

$$J_T(\theta + \delta) \leq J_T(\theta) + \langle \nabla J_T(\theta), \delta\rangle + \frac{L_T}{2}\|\delta\|^2. \tag{13}$$

*In particular, taking $\theta = \theta_t^{\mathrm{pre}}$ and $\delta = \delta(t, w) := \theta_0(t, w) - \theta_t^{\mathrm{pre}}$ yields*

$$J_T\big(\theta_0(t, w)\big) \leq J_T\big(\theta_t^{\mathrm{pre}}\big) + \big\langle \nabla J_T\big(\theta_t^{\mathrm{pre}}\big), \delta(t, w)\big\rangle + \frac{L_T}{2}\|\delta(t, w)\|^2. \tag{14}$$

*Consequently, a sufficient condition for midtraining to decrease the SFT loss at initialization, $J_T(\theta_0(t, w)) \leq J_T(\theta_t^{\mathrm{pre}})$, is*

$$\big\langle \nabla J_T\big(\theta_t^{\mathrm{pre}}\big), \delta(t, w)\big\rangle \leq -\frac{L_T}{2}\|\delta(t, w)\|^2. \tag{15}$$

*Proof.* This is a direct application of the descent lemma (Theorem A.3) with $f = J_T$, $x = \theta$, and $y = \theta + \delta$:

$$J_T(\theta + \delta) \leq J_T(\theta) + \langle \nabla J_T(\theta), (\theta + \delta) - \theta\rangle + \frac{L_T}{2}\|(\theta + \delta) - \theta\|^2$$

$$= J_T(\theta) + \langle \nabla J_T(\theta), \delta\rangle + \frac{L_T}{2}\|\delta\|^2.$$

Substituting $\theta = \theta_t^{\mathrm{pre}}$ and $\delta = \delta(t, w)$ gives Equation 14. Finally, $J_T(\theta_0(t, w)) \leq J_T(\theta_t^{\mathrm{pre}})$ holds whenever the sum of the linear and quadratic terms in Equation 14 is nonpositive, i.e., whenever Equation 15 holds. $\square$

**Lemma A.6** (Bounding the midtraining displacement). *Suppose midtraining runs for $m$ steps starting from $\theta_t^{\text{pre}}$ and produces $\theta_0(t, w)$. Let $\{\varphi_u\}_{u=0}^m$ be the midtraining iterates with*

$$\varphi_0 = \theta_t^{\text{pre}}, \qquad \varphi_m = \theta_0(t, w), \qquad \delta(t, w) = \varphi_m - \varphi_0.$$

*Assume the midtraining updates have the form*

$$\varphi_{u+1} = \varphi_u - \alpha\, s(t + u)\, g_u, \qquad u = 0, \dots, m - 1, \tag{16}$$

*where $\alpha > 0$ is the midtraining step size, $s(\cdot) \in [0, 1]$ is nonincreasing (representing a loss of plasticity or other proxy for diminishing leverage of later updates compared to earlier ones), and $g_u$ is the (stochastic) gradient used at step $u$. Define the cumulative plasticity over the block*

$$S(t) := \sum_{u=0}^{m-1} s(t + u).$$

*If $\|g_u\| \le G(w)$ for all $u$ in the block, then*

$$\|\delta(t, w)\| \le \alpha\, G(w)\, S(t), \tag{17}$$

*and hence*

$$\frac{L_T}{2} \|\delta(t, w)\|^2 \le \frac{L_T}{2}\, \alpha^2\, G(w)^2\, S(t)^2. \tag{18}$$

*Proof.* Summing the increments in Equation 16 yields

$$\delta(t, w) = \varphi_m - \varphi_0 = \sum_{u=0}^{m-1} (\varphi_{u+1} - \varphi_u) = -\alpha \sum_{u=0}^{m-1} s(t + u)\, g_u.$$

Taking norms and applying the triangle inequality,

$$\|\delta(t, w)\| = \left\| -\alpha \sum_{u=0}^{m-1} s(t + u)\, g_u \right\| \le \alpha \sum_{u=0}^{m-1} s(t + u)\, \|g_u\|$$

$$\le \alpha\, G(w) \sum_{u=0}^{m-1} s(t + u) = \alpha\, G(w)\, S(t),$$

which proves Equation 17. Squaring and multiplying by $L_T/2$ gives Equation 18. $\square$

**Plugging into the bound.** Applying Equation 12 with $\theta_0 = \theta_0(t, w)$ gives

$$\Delta_P(K) \le -\eta \sum_{k=0}^{K-1} \langle \nabla J_P(\theta_k), \nabla J_T(\theta_k) \rangle + L_P \eta \Big( J_T(\theta_0(t, w)) - J_T^* \Big)$$

By Lemma A.5 (descent lemma applied to $J_T$ at $\theta_t^{\text{pre}}$ with displacement $\delta(t, w)$),

$$J_T(\theta_0(t, w)) \le J_T(\theta_t^{\text{pre}}) + \langle \nabla J_T(\theta_t^{\text{pre}}), \delta(t, w) \rangle + \frac{L_T}{2} \|\delta(t, w)\|^2$$

By Lemma A.6, $\|\delta(t, w)\| \le \alpha\, G(w)\, S(t)$, hence

$$\frac{L_T}{2} \|\delta(t, w)\|^2 \le \frac{L_T}{2} \alpha^2 G(w)^2 S(t)^2$$

Combining these inequalities yields

$$\Delta_P(K) \le -\eta \sum_{k=0}^{K-1} \langle \nabla J_P(\theta_k), \nabla J_T(\theta_k) \rangle$$
$$+ L_P \eta \Big[ (J_T(\theta_t^{\text{pre}}) - J_T^*) + \langle \nabla J_T(\theta_t^{\text{pre}}), \delta(t, w) \rangle + \frac{L_T}{2} \alpha^2 G(w)^2 S(t)^2 \Big]. \tag{19}$$

This expression also motivates the interaction between midtraining start time and mixture weight that we studied in section 6. Later start times reduce the remaining plasticity term, limiting how much the initialization can be improved before posttraining. Higher mixture weights increase the strength of the target-domain update. The bound therefore suggests that late introduction and high mixture weight can interact unfavorably: when little plasticity remains, aggressive specialization may increase forgetting without producing a sufficiently better posttraining initialization.

## B. Pretraining Settings

We pretrained models from scratch on the C4 dataset. All four models were trained for 128B tokens or approximately 61k steps, with very similar settings (documented in Table 5). L40S GPUs were used for all pretraining and midtraining runs. Models were trained with the LitGPT library (Lightning AI, 2023).

*Table 5.* Core pretraining hyperparameters for Pythia-70M, 160M, 410M, and 1B.

| Hyperparameter | 70M | 160M | 410M | 1B |
|---|---|---|---|---|
| Global batch size | 1024 | 1024 | 1024 | 1024 |
| Micro batch size | 16 | 16 | 8 | 4 |
| LR schedule | Cosine w/ 10% warmup | Cosine w/ 10% warmup | Cosine w/ 10% warmup | Cosine w/ 10% warmup |
| Max LR | $3 \times 10^{-4}$ | $3 \times 10^{-4}$ | $3 \times 10^{-4}$ | $3 \times 10^{-4}$ |
| Min LR | $1 \times 10^{-6}$ | $1 \times 10^{-6}$ | $1 \times 10^{-6}$ | $1 \times 10^{-6}$ |
| Optimizer | AdamW | AdamW | AdamW | AdamW |
| Betas | (0.9, 0.95) | (0.9, 0.95) | (0.9, 0.95) | (0.9, 0.95) |
| Weight decay | 0.1 | 0.1 | 0.1 | 0.1 |
| Precision | BF16 | BF16 | BF16 | BF16 |
| Num GPUs | 4 | 4 | 8 | 8 |

## C. Posttraining Settings

We fine-tuned all models on four downstream datasets: Pycode (our 5K-sample subset of CodeSearchNet-Python), GSM8K (7.5K math problems), LIMA (1K instruction examples), and SciQ (13.7K science questions). For GSM8K only, the prompt/question portion was masked during loss; for the others the loss was computed over the full sequence. A summary of the datasets is given in Table 6. All runs used a cosine learning rate schedule with 10% linear warmup, trained for 4 epochs, global batch size 64, and micro-batch size 16 for 70M/160M (8 for 410M). Peak learning rates were selected by grid search on the base pretrained checkpoint before midtraining, and the LR grid is given in Table 7. Selected LRs for the final checkpoint of each model size are given in Table 8.

*Table 6.* Finetuning datasets.

| Dataset | # Train Samples | Prompt masked |
|---|---|---|
| Pycode (CodeSearchNet-Python subset) | 5,000 | No |
| GSM8K | 7,500 | Yes |
| LIMA | 1,000 | No |
| SciQ | 13,679 | Yes |

## D. Dataset Similarity Matrix

We compute dataset similarity using surface-level token statistics after initial experimentation with embedding models gave implausible results for code datasets' similarities to other natural language datasets. For each pair of pretrain/midtrain

*Table 7.* Grid of candidate peak learning rates swept during tuning.

**LR grid**

```
4e-6, 8e-6, 1e-5, 2e-5, 4e-5, 5e-5, 6e-5, 7e-5, 8e-5, 9e-5, 1e-4, 1.2e-4, 1.4e-4, 1.6e-4,
1.8e-4, 2e-4, 2.4e-4, 4e-4, 5e-4, 6e-4, 8e-4, 1e-3, 2e-3, 3e-3, 4e-3, 6e-3
```

*Table 8.* Selected peak learning rates for fine-tuning (cosine schedule with 10% warmup).

| Dataset | 70M | 160M | 410M | 1B |
|---|---|---|---|---|
| GSM8K | 8e-4 | 4e-4 | 4e-4 | 7e-05 |
| LIMA | 1.2e-4 | 5e-5 | 5e-5 | 2e-05 |
| Pycode | 1e-3 | 5e-4 | 4e-4 | 9e-05 |
| SciQ | 8e-4 | 2.4e-4 | 6e-4 | 9e-05 |

and downstream datasets, we sample $\max(\text{dataset\_size}, 10{,}000)$ examples. Midtrain mixes are simulated by their actual compositions (e.g., Starcoder is treated as 20% Starcoder + 80% C4). From the (possibly mixed) texts we build unigram frequency vectors at a token level, normalize to probabilities, and compute: vocabulary Jaccard, overlap ratio, token-frequency cosine similarity, and a Jensen–Shannon-based similarity. These are combined as

$$\text{Combined} = 0.4 \cdot \text{cosine} + 0.3 \cdot \text{Jaccard} + 0.3 \cdot \text{JS\_similarity},$$

and used to fill the similarity matrix (diagonal entries are 1). This mixture-aware score reflects both specialty content and dilution by C4. Figure 7 shows the resulting similarity matrix between pre/midtrain datasets and SFT datasets.

## E. SFT in-domain loss and C4 Losses after Finetuning for 70m and 160m models

Table 9 depicts validation losses as well as C4 validation losses after finetuning on each SFT dataset, for the 70m and 160m models.

*Table 9.* SFT and C4 validation losses for 70M, 160M, and 410M models across downstream datasets and midtraining mixtures, averaged across 5 seeds for each SFT dataset. Bold values indicate best performance within each dataset and model size combination.

| Model Size | Downstream Dataset | Midtrain Mix | SFT Val Loss | C4 Val Loss |
|---|---|---|---|---|
| 70m | Pycode | C4 | 2.656 | 6.152 |
| | | Starcoder (20%) | **2.504** | **6.032** |
| | | Math (12%) | 2.603 | 6.116 |
| | | FLAN (5%) | 2.802 | 6.400 |
| | | KnowledgeQA (20%) | 2.628 | 6.117 |
| | | DCLM (20%) | 2.584 | 6.052 |
| | GSM8K | C4 | 1.384 | 6.353 |
| | | Starcoder (20%) | 1.353 | **6.317** |
| | | Math (12%) | **1.339** | 6.358 |
| | | FLAN (5%) | 1.368 | 6.352 |
| | | KnowledgeQA (20%) | 1.367 | 6.376 |
| | | DCLM (20%) | 1.368 | 6.352 |
| | LIMA | C4 | 4.333 | 4.124 |
| | | Starcoder (20%) | 4.346 | 4.136 |
| | | Math (12%) | 4.362 | 4.146 |
| | | FLAN (5%) | 4.342 | 4.110 |
| | | KnowledgeQA (20%) | **4.290** | 4.110 |
| | | DCLM (20%) | 4.324 | **4.097** |
| | SciQ | C4 | 3.159 | 7.703 |
| | | Starcoder (20%) | 3.187 | 7.804 |
| | | Math (12%) | 3.187 | 7.971 |
| | | FLAN (5%) | 3.161 | 7.888 |

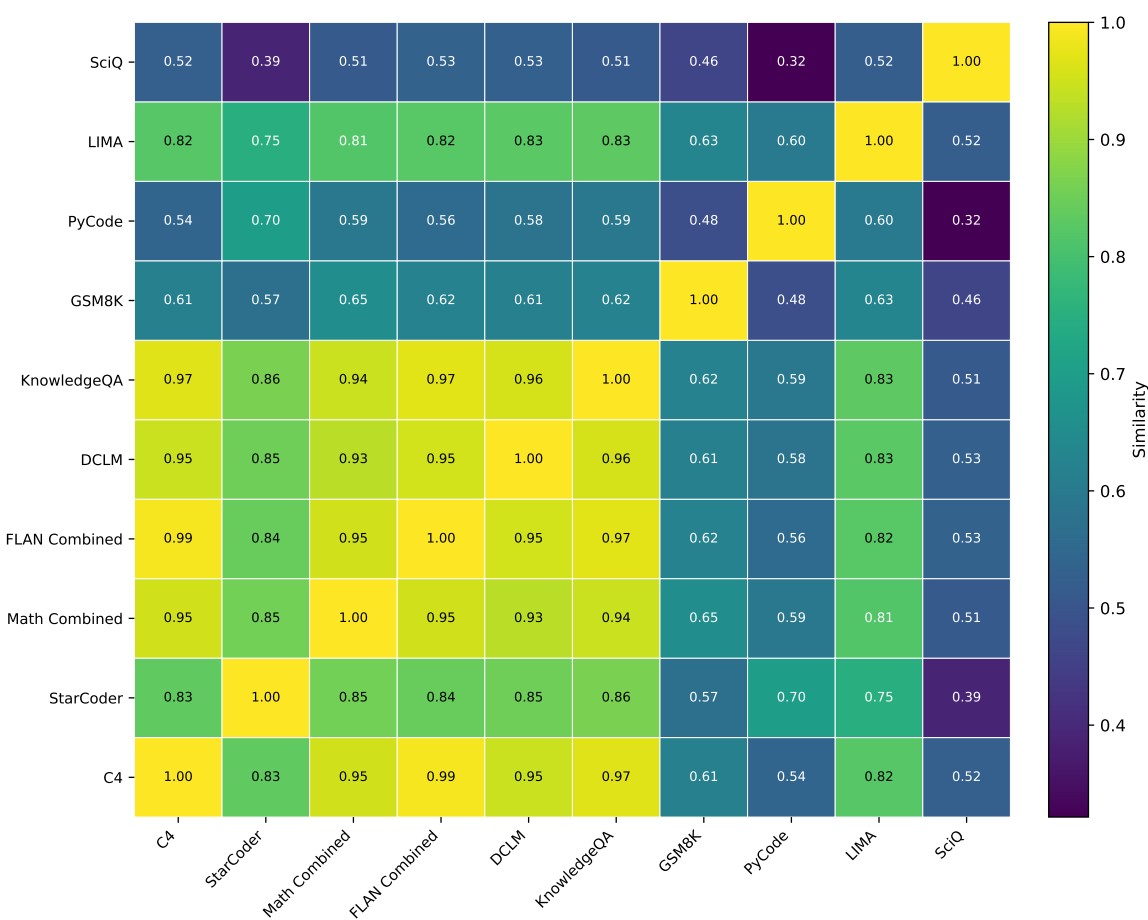

*Figure 7.* Token-based similarity matrix for pre/midtrain and SFT datasets. Note that these midtrain datasets are corrected for mix weight in this matrix.

| Model Size | Downstream Dataset | Midtrain Mix | SFT Val Loss | C4 Val Loss |
|---|---|---|---|---|
| | | KnowledgeQA (20%) | **3.142** | **7.567** |
| | | DCLM (20%) | 3.147 | 7.703 |
| 160m | Pycode | C4 | 2.314 | 5.254 |
| | | Starcoder (20%) | **2.134** | **5.079** |
| | | Math (12%) | 2.332 | 5.277 |
| | | FLAN (5%) | 2.318 | 5.257 |
| | | KnowledgeQA (20%) | 2.306 | 5.232 |
| | | DCLM (20%) | 2.305 | 5.215 |
| | GSM8K | C4 | 1.163 | 5.308 |
| | | Starcoder (20%) | 1.134 | 5.315 |
| | | Math (12%) | **1.114** | **5.230** |
| | | FLAN (5%) | 1.152 | 5.299 |
| | | KnowledgeQA (20%) | 1.145 | 5.287 |
| | | DCLM (20%) | 1.149 | 5.303 |
| | LIMA | C4 | 3.828 | 3.578 |
| | | Starcoder (20%) | 3.810 | 3.581 |
| | | Math (12%) | 3.795 | 3.569 |
| | | FLAN (5%) | 3.836 | 3.559 |
| | | KnowledgeQA (20%) | **3.736** | 3.560 |
| | | DCLM (20%) | 3.792 | **3.549** |
| | SciQ | C4 | 2.705 | 4.423 |
| | | Starcoder (20%) | 2.728 | 4.474 |
| | | Math (12%) | 2.740 | 4.343 |
| | | FLAN (5%) | 2.708 | 4.427 |
| | | KnowledgeQA (20%) | 2.673 | **4.159** |
| | | DCLM (20%) | **2.671** | 4.377 |
| 410m | Pycode | C4 | 2.151 | 5.032 |
| | | Starcoder (20%) | **1.971** | **4.608** |
| | | Math (12%) | 2.159 | 5.109 |
| | | FLAN (5%) | 2.152 | 4.920 |
| | | KnowledgeQA (20%) | 2.151 | 5.052 |
| | | DCLM (20%) | 2.159 | 5.109 |
| | GSM8K | C4 | 1.043 | 4.952 |
| | | Starcoder (20%) | 1.029 | 4.923 |
| | | Math (12%) | **1.004** | **4.872** |
| | | FLAN (5%) | 1.050 | 5.089 |
| | | KnowledgeQA (20%) | 1.043 | 4.928 |
| | | DCLM (20%) | 1.056 | 5.052 |
| | LIMA | C4 | 3.446 | 3.178 |
| | | Starcoder (20%) | **3.403** | **3.162** |
| | | Math (12%) | 3.452 | 3.180 |
| | | FLAN (5%) | 3.471 | 3.175 |
| | | KnowledgeQA (20%) | 3.468 | 3.173 |
| | | DCLM (20%) | 3.463 | 3.170 |
| | SciQ | C4 | 2.247 | 3.646 |
| | | Starcoder (20%) | **2.223** | 3.593 |
| | | Math (12%) | 2.255 | 3.610 |
| | | FLAN (5%) | 2.233 | 3.581 |
| | | KnowledgeQA (20%) | 2.226 | **3.541** |
| | | DCLM (20%) | 2.240 | 3.647 |

## F. Representative training loss curves for midtrained vs. base models

Figure 8 shows a representative training loss curve for a midtrained model when its domain is aligned to SFT data.

## G. Additional CKA results on APPS

Figure 9 and Figure 10 display the CKA layer similarity for 160m and 410m models.

## H. CKA results on C4

Figure 11, Figure 12, and Figure 13 show the CKA layer similarity for all model sizes with C4 as a probe.

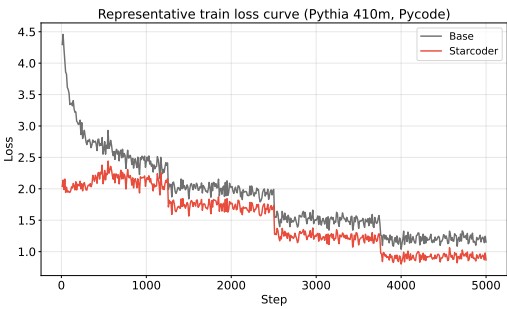

*Figure 8.* Representative training loss curve for a midtrained model and base model on Pycode, for Pythia-410m. The midtrained model starts with a lower training loss, and maintains a slight gap throughout training.

*Figure 9.* CKA layer analysis for Pythia-160M with APPS as a probe.

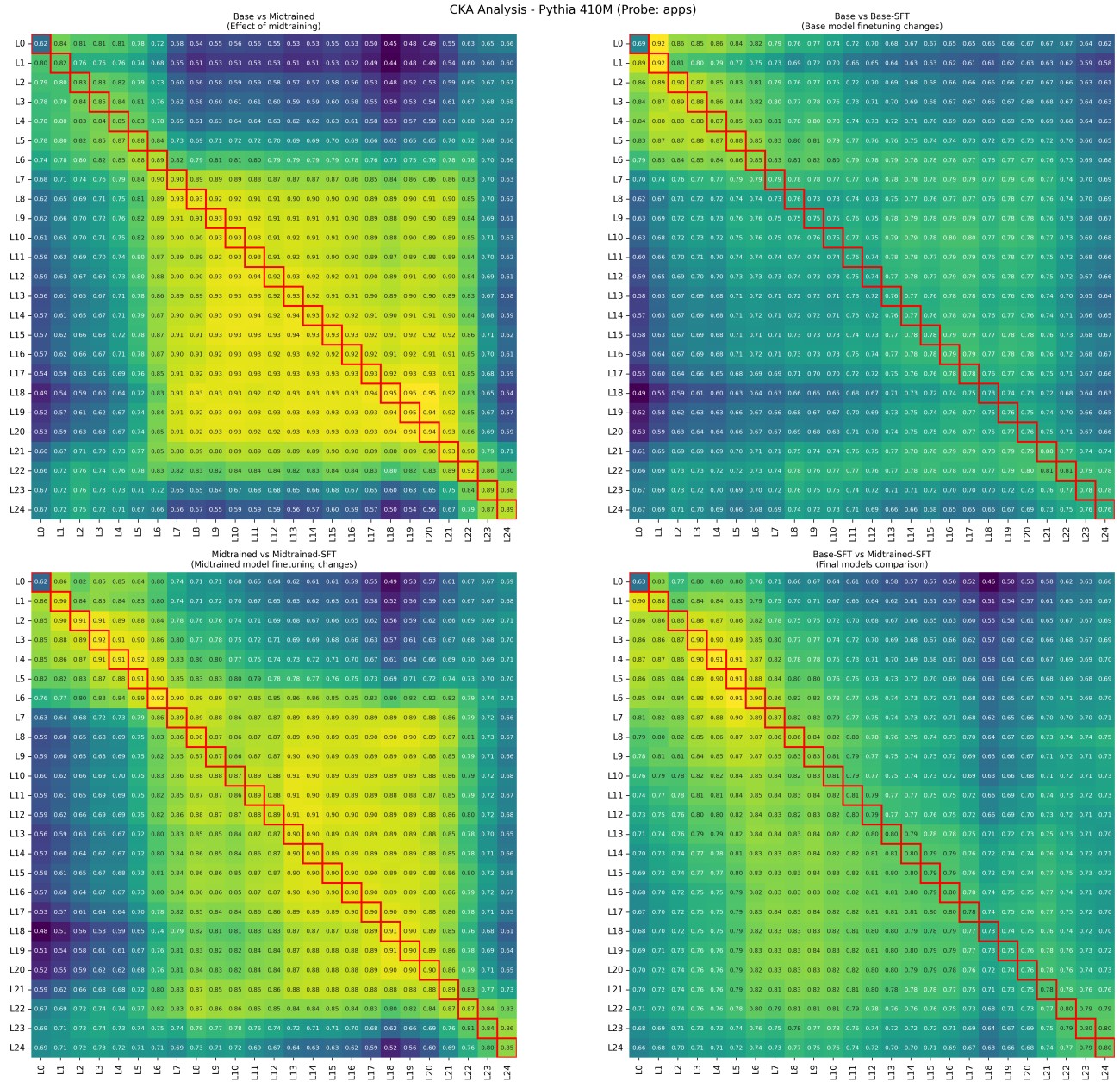

*Figure 10.* CKA layer analysis for Pythia-410M with APPS as a probe.

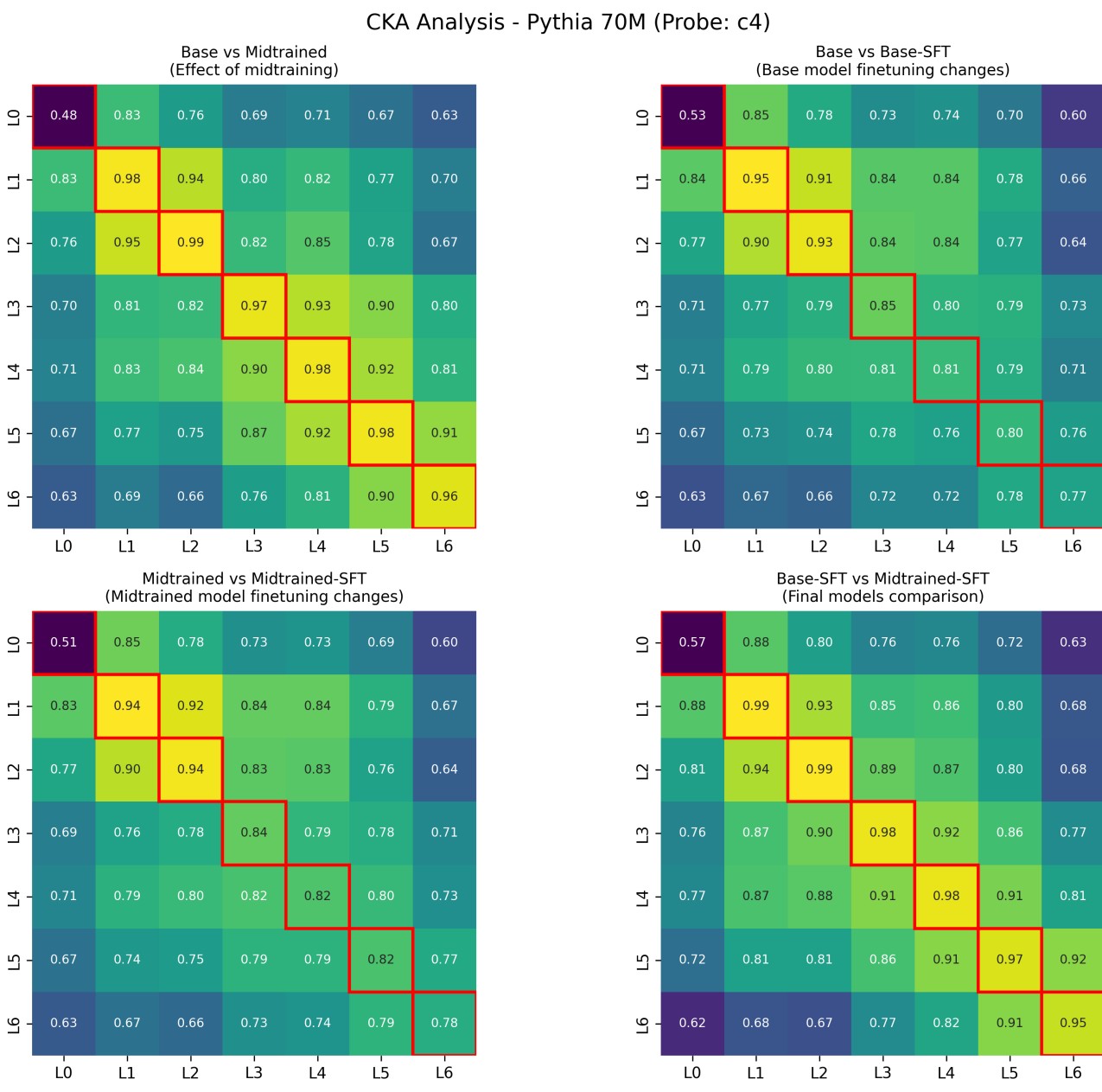

*Figure 11.* CKA layer analysis for Pythia-70M with C4 as a probe.

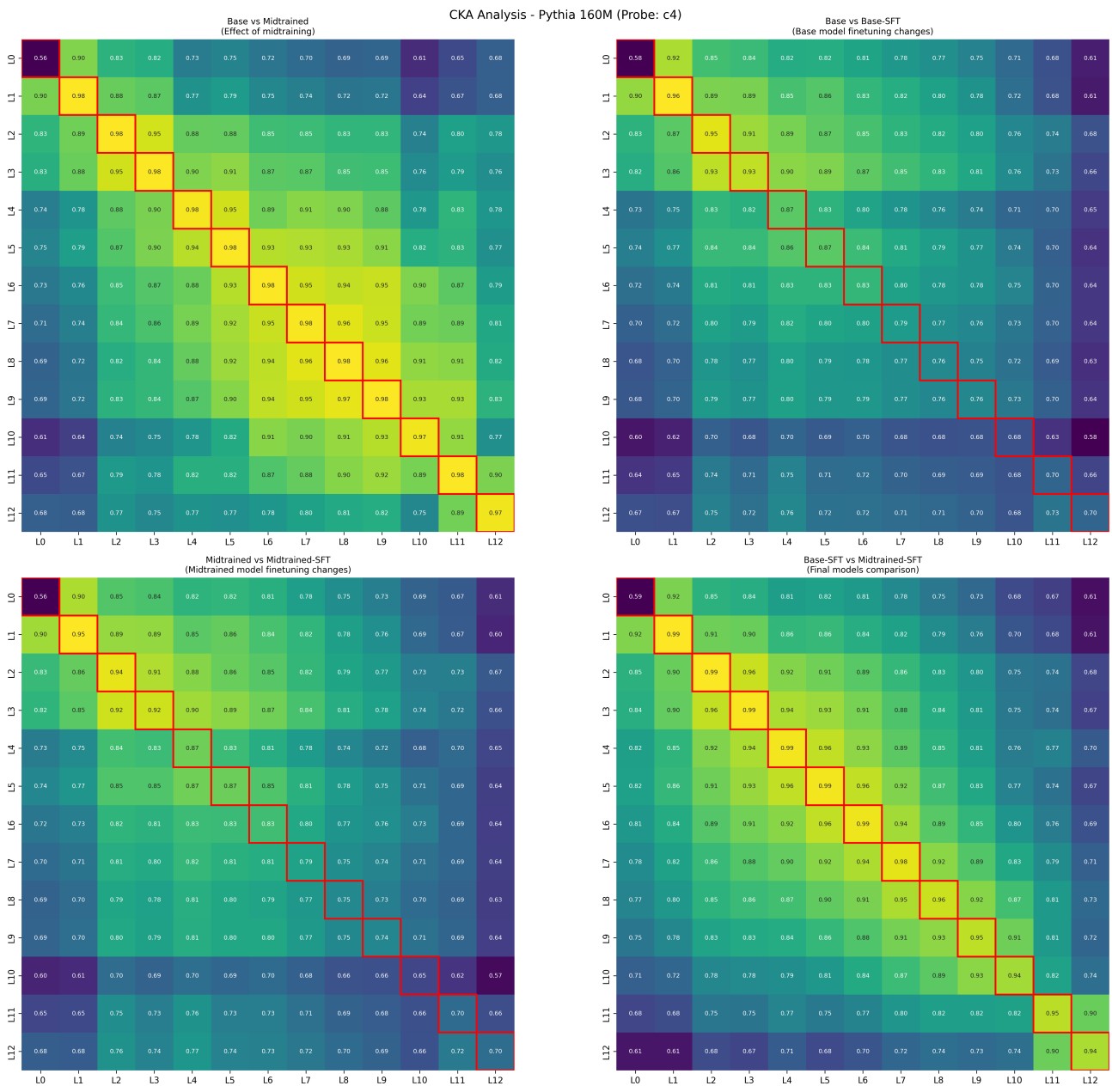

*Figure 12.* CKA layer analysis for Pythia-160M with C4 as a probe.

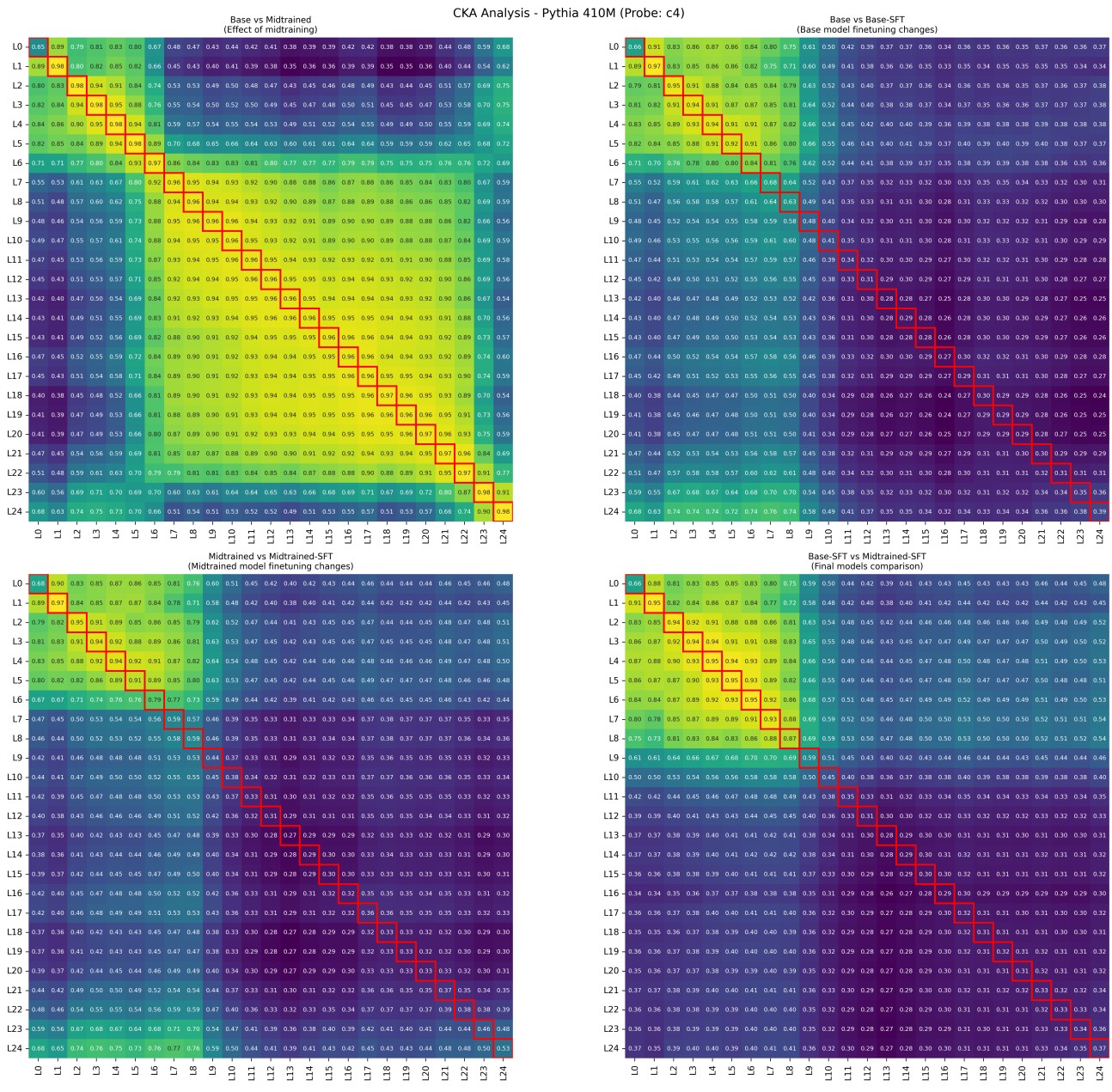

*Figure 13.* CKA layer analysis for Pythia-410M with C4 as a probe.

# I. Statement on LLM Usage

Large language models (LLMs) were used to assist with refining writing in this submission, including summarizing paragraphs in order to shorten the submission, correcting grammar, and giving suggestions to improve organization. LLMs were not used in the ideation process and analyses and experimental setups were designed fully by the authors. Copilot and other coding agents were used to generate some utility scripts in the process of coding.

