# OpenReview forum: "Midtraining Bridges Pretraining and Posttraining Distributions"
_ICML.cc/2026/Conference — ICML 2026 spotlight_

### Official Review · Reviewer_MHUz · 2026-03-07

**Soundness:** 3
**Presentation:** 3
**Significance:** 2
**Originality:** 2
**Overall Recommendation:** 4
**Confidence:** 4

**Summary:**

This paper presents a systematic empirical study of midtraining, the practice of inserting an intermediate training phase between pretraining and posttraining that mixes specialized data with general pretraining data. They find that midtraining benefits are domain-specific, correlate with a token-level proximity metric, outperform continued pretraining, and exhibit a strong interaction between timing and mixture weight.

**Compliance With Llm Reviewing Policy:**

Affirmed.

**Final Justification:**

Overall, the rebuttal provided supportive evidence. Although the core idea of distributional bridging through intermediate training is known in prior work, such as domain-adaptive pretraining (Gururangan et al., 2020) and pre-finetuning for distribution shift (Kang et al., 2024), which the authors themselves acknowledge as similar in spirit, as well as intermediate task transfer (Phang et al., 2018, "Sentence Encoders on STILTs: Supplementary Training on Intermediate Labeled-data Tasks"), I appreciate the empirical execution for LLMs and the timing-weight interaction result, which I find most interesting.

I recommend weak accept.

**Key Questions For Authors:**

Q1 Have you measured actual gradient alignment (term A in your bound) empirically? This would directly test the theory rather than relying on the proxy of loss values.

Q2 Does resetting or rewinding the learning rate schedule at the midtraining start point change the timing-weight interaction? This would help distinguish the "plasticity window" hypothesis from a simpler "learning rate is too low" explanation.

**Limitations:**

It has an Impact Statement after the Conclusion, which briefly touches on compute reduction and data governance, but limitations are only mentioned in a single sentence at the end of the Conclusion ("it will be important to test whether these trends persist at larger scales and across a broader range of domains"). I concur.

**Strengths And Weaknesses:**

Strengths
- S1. Clear writing and framing. The "distributional bridging" framing is intuitive and well-motivated. The findings are stated concisely and are easy to extract.
- S2. The timing-weight interaction is interesting. The finding that early introduction supports aggressive mixture weights while late introduction makes them harmful is non-obvious and practically important. Personally, I think this is the paper's strongest and most novel finding.

Weaknesses
- W1. Limited novelty in several findings. Finding 1 (midtraining benefits are domain-specific) is expected. Of course code data helps code tasks. Finding 3 (mixing general data is better than pure specialized data) is well-established in the continual learning literature under various names (rehearsal, replay, data mixing). The proximity advantage metric (Finding 2) is a straightforward token-overlap measure that provides directional but not precise predictions. The paper's novelty rests heavily on Finding 4 (timing-weight interaction), which is strong, and Finding 5 (CKA analysis), which is underdeveloped.
- W2. CKA analysis (Section 7) is preliminary. The observation that midtrained models show slightly higher CKA similarity in the final layer after fine-tuning is interesting but underdeveloped. The bottom-right panel of Figure 6 shows that final fine-tuned models are very similar regardless of midtraining (CKA > 0.96 in final layers), which somewhat undermines the narrative that midtraining fundamentally changes the optimization trajectory. More analysis is needed. What about gradient alignment measurements, which the theory emphasizes? What about loss landscape visualization?
- W3. The paper should discuss whether the learning rate schedule interacts with midtraining timing. The cosine schedule means the learning rate is decaying throughout training, which could partially explain why late introduction with high mixture weights fails. It might be a learning rate issue, not just a "plasticity" issue.

---

> ### Author Rebuttal · Authors · 2026-03-31
>
> We thank you for the detailed and constructive feedback. We address your concerns below:
>
> **W1: Limited novelty in Findings 1-3.**
>
>  We respectfully push back on this characterization, and want to draw a distinction between results that are unsurprising in hindsight and results that are not novel. Prior to this work, there was no systematic controlled study of midtraining as the practice was mostly documented in LM tech reports, which by their nature report working recipes rather than the full experimental landscape. We believe that it is important to systematically study this training stage, even if some of the initial results are intuitively expected. We will revise the introduction to make this "systematic first study" framing more explicit, and to clearly distinguish our contribution from what can be inferred from existing tech reports.
>
> **W2: CKA analysis is preliminary**
>
> We push back on the interpretation that high final-layer CKA similarity undermines the midtraining narrative. Prior work  (https://arxiv.org/abs/2202.10054) shows that one signature of feature distortion during finetuning and subsequent forgetting of pretraining data is heavy changes to the final layer, which is precisely the pattern we observe in the base model but not the midtrained model. The fact that both models converge to similar final representations is not a contradiction, as it shows they reach the same destination via different paths, with midtraining making that path less disruptive. However, we agree the CKA section may not be as strong and will restructure the discussion to make this argument explicit, and add gradient alignment measurements along the posttraining trajectory to directly test term (A) in our bound.
>
> **W3/Q2: Learning rate schedule as a confound**
>
> If the failure of late x high mixture weight were purely a consequence of low LR, we would expect all late-introduction conditions to fail similarly regardless of mixture weight. Instead, late introduction with low mixture weight still performs reasonably and it is specifically the late x high weight combination that degrades performance. An LR-based explanation does not predict this interaction. We agree that LR schedules are important to consider though, and will also run an ablation with the WSD learning rate schedule in the camera ready, which maintains a high constant learning rate throughout most of training.
>
> **Q1: Empirical gradient alignment measurements**
>
> We agree that this would be a valuable addition to the paper. In order to quickly quantify this, we measured the grad norm of the base and code-midtrained model on the pycode SFT dataset across 50 samples selected with 3 different random seeds, which quantifies the adaptation effort needed during finetuning. This was found to be 2.78 (+- 0.0098) for the base model and 0.733 (+- 0.016) for the midtrained model, which agrees with our theoretical framing. That said, we will also work on getting more concrete gradient alignment results during training and will add an expanded section on this in the revised paper.

---

> > ### Author Rebuttal · Reviewer_MHUz · 2026-03-31
> >
> > I thank the authors for the detailed response. A few remaining questions.
> >
> > On W1 (Novelty): I accept the distinction between "unsurprising in hindsight" and "not novel," and the framing as a systematic controlled study is partially fair. That said, prior work such as FRAME (Zhang et al., ACL 2025) and SmolLM2 (Allal et al., 2025) has already conducted controlled multi-stage training experiments, so the "first systematic study" claim should be qualified more carefully.
> >
> > On W2 (CKA): The reference to prior work on feature distortion as a signature of forgetting is helpful context. However, the core observation remains that final-layer CKA exceeds 0.96 for both conditions after SFT, which makes it hard to argue that midtraining produces a meaningfully different outcome at the representation level. The "same destination, different paths" interpretation is plausible but needs trajectory-level evidence to be convincing. My original review also suggested loss landscape visualization as one way to substantiate this, which was not addressed.
> >
> > On Q1 (Gradient alignment): The grad norm comparison (2.78 vs 0.733) is informative but seems corresponds to the effort term in the bound. My question was about gradient alignment, the other term in the bound, which concerns directional agreement between the gradient and the displacement toward the optimum. These are different quantities, and the theory treats them as separate contributions to the forgetting bound. Could the authors clarify whether they plan to measure directional alignment specifically, or whether they consider grad norm a sufficient proxy?

---

> > > ### Author Response · Authors · 2026-04-07
> > >
> > > Thank you again for the detailed feedback and push to make distinctions sharper. We address each of the remaining points below:
> > >
> > > **W1: novelty/first systematic study**
> > >
> > > We can further qualify this wording in the draft. Our intent was to emphasize the controlled comparison of midtraining data domain, distributional proximity, continued pretraining, and the timing x mixture interaction within one unified setup, not to suggest that prior work had not performed any controlled multi-stage training experiments. We do discuss past works conducting multi-stage pretraining in our related works as well.
> > >
> > > **W2: CKA/ same destination different paths**
> > >
> > > To clarify, our intended claim was not that midtraining should lead to a large final representational difference after SFT. Rather, the point of the CKA section was that midtraining may alter the initial $\theta_0$ even if the final representations end up quite similar. So we agree that final-layer CKA values above 0.96 make it hard to argue for a large endpoint difference, but that is not the claim we meant to make.
> > >
> > > To better ground this transition-level interpretation, we added two initialization-level diagnostics at the pre-SFT checkpoint:
> > >
> > > - Pre-SFT target loss vs. pre-SFT c4 loss: in the matched-domain settings, midtraining substantially improves the target initialization while leaving held out C4 nearly unchanged:
> > >
> > > | Model | Base PyCode | Code-mid PyCode | Δ(PyCode) | Base C4 | Code-mid C4 | Base GSM8K | Math-mid GSM8K | Δ(GSM8K) | Base C4 | Math-mid C4 |
> > > |---|---:|---:|---:|---:|---:|---:|---:|---:|---:|---:|
> > > | 70M  | 5.441 | 3.248 | -2.193 | 4.121 | 4.160 | 4.100 | 2.653 | -1.447 | 4.121 | 4.162 |
> > > | 160M | 4.798 | 2.561 | -2.237 | 3.634 | 3.664 | 3.590 | 1.918 | -1.672 | 3.634 | 3.646 |
> > > | 410M | 4.459 | 2.136 | -2.323 | 3.256 | 3.254 | 3.163 | 1.429 | -1.734 | 3.256 | 3.265 |
> > > | 1B   | 4.151 | 1.948 | -2.203 | 3.002 | 3.032 | 2.886 | 1.301 | -1.585 | 3.002 | 2.998 |
> > >
> > > - Posttraining only forgetting: using pre-SFT C4 loss as $J_P(\theta_0)$ we can directly compute the posttraining forgetting quality $J_P(\theta_K) - J_P(\theta_0)$ which is the quantity the theory sketch is intended to describe:
> > >
> > > | Setting | Pre-SFT C4 | Post-SFT C4 | Posttraining forgetting Δ |
> > > |---|---:|---:|---:|
> > > | 1B base → PyCode SFT | 3.002 | 3.075 | 0.073 |
> > > | 1B code-midtrained → PyCode SFT | 3.032 | 3.070 | 0.038 |
> > > | 1B base → GSM8K SFT | 3.002 | 3.134 | 0.132 |
> > > | 1B math-midtrained → GSM8K SFT | 2.998 | 3.018 | 0.020 |
> > >
> > > The intended point was that the main empirical difference is at the initialization for posttraining, not necessarily a large final endpoint separation. We will revise the CKA discussion to make this clear.
> > >
> > > On your suggestion of a loss landscape visualization: we agree this is a reasonable way to probe the “same destination, different paths” intuition. We had explored local loss-landscapes (on an open midtrained model, OLMo-2 7B) in an earlier version of the project and found them supportive of the bridging intuition, but these analyses were not carried through into the final experimental setup after the project pivoted. Given rebuttal time constraints, we prioritized adding the more direct initialization-level diagnostics above. We agree that a loss-landscape view would be a useful addition, and if space permits in the revision we will add an illustrative loss landscape figure.
> > >
> > > **Q1: Gradient alignment vs gradnorm**
> > >
> > > To address this concern more directly, we added an empirical directional-alignment analysis. Raw gradient-cosine estimates were somewhat noisy at our model scales, we used a finite-difference probe where we take a small step in the target-SFT gradient direction and measure the induced change in held out C4 loss. This is a measurement of the one-step forgetting effect on the left hand side of our bound, i.e. $J_P(\theta_{t+1}) - J_P({\theta_t})$. We want mid ΔJ_C4 to be lower than base ΔJ_C4 in the table below, and a negative number means that moving in the target update direction of the SFT dataset would lower C4 loss as well.
> > >
> > > | Model | base ΔJ_C4 | mid ΔJ_C4 | ratio (mid/base) |
> > > |---|---:|---:|---:|
> > > | 70m_code  | +3.1e-4 | -2.2e-5 | -0.07× |
> > > | 70m_math  | +1.2e-3 | +2.4e-3 | +1.89× |
> > > | 160m_code | +6.0e-5 | -4.3e-5 | -0.71× |
> > > | 160m_math | +2.0e-5 | -5.3e-5 | -2.68× |
> > > | 410m_code | +1.2e-4 | +2.1e-5 | +0.18× |
> > > | 410m_math | +6.7e-5 | +4.1e-6 | +0.06× |
> > > | 1b_math   | +7.2e-6 | -1.6e-5 | -2.26× |
> > > | 1b_code   | +8.6e-5 | +3.0e-5 | +0.35× |
> > >
> > > In the main successful regimes, this target step causes substantially less immediate C4 degradation from the midtrained initialization than from the base initialization. This is qualitatively consistent with the alignment/conflict term in the theory: in the successful midtraining settings, the target update direction is less harmful to the pretraining objective at initialization. The notable exception is 70M-math, which is unsurprising as the C4 loss for 70M-math is actually not lower in the main results (see Table 8).

---

### Official Review · Reviewer_gFQ8 · 2026-03-12

**Soundness:** 4
**Presentation:** 4
**Significance:** 3
**Originality:** 3
**Overall Recommendation:** 6
**Confidence:** 4

**Summary:**

This paper investigates the effect of pretraining on final model performance (in terms of target distribution loss and pretraining loss, which measures potential forgetting). The authors motivate with a high-level theoretical framework that midtraining might serve to bridge the pre- and posttraining distributions while representations are yet malleable, yielding improved performance on the target data with minimal forgetting. The authors undermine this view with a series of experiments across a range of model scales, providing convincing evidence. The authors also propose a simple and cheap measure, proximity advantage, to gauge whether a midtraining distribution can constitue an effective data bridge. Further, the authors provide guidance backed by empirical observations on effective timing and weighting of midtraining data.

**Compliance With Llm Reviewing Policy:**

Affirmed.

**Final Justification:**

I believe this work to be of great interest to the community, as midtraining is currently understudied. I continue to recommend acceptance.

**Key Questions For Authors:**

Please refer to the weakness section above.

**Limitations:**

I believe limitations in terms of overall model architecture / pretraining recipes could be discussed. Do the authors believe the results would hold across model families (e.g., are there observations in related literature to support that)?

**Strengths And Weaknesses:**

# Strengths
1. This work considers an important yet scarcely studied part of modern LLM training pipelines: midtraining. The results are expected to be of great interest to the community, and the actionable insights are valuable contributions to the field.
2. The paper provides simple tools (proximity advantage, guidance on timing and weighting of midtraining data) that will be useful to model developers.
3. The authors provide a simple and intuitive interpretation of their findings, supported by a clean theoretical analysis. The findings are well explained and put into context with observations in related literature.
4. The analysis is rigorous; the authors took care to run hyperparameter sweeps, evaluate multiple seeds, and report significance levels.
5. The manuscript is extremely well structured, easy to follow, and very well written. The presentation of a simple yet useful definition of midtraining is a welcome inclusion. Figures are readable, well organized, and to the point.

# Weaknesses
1. It remains somewhat unclear how the specific formulation of the proximity term (i.e., the weights for cosine similarity, vocabulary Jaccard, and JS-similarity) came to be (App. D), and whether results (in particular, the correlation with proximity advantage in Fig. 4) are robust to slightly different formulations. It would be useful to consider the interplay of these terms and evaluate their usefulness in isolation (e.g., if proximity advantage in terms of each individual term (like cosine similarity) is still strongly correlated with relative improvement, is it even necessary to combine all terms? Conversely, if one term is less correlated, is it meaningfully contributing to the overall proximity?).
2. I'm missing an explanation of how the weights for the default midtraining mixtures are chosen (e.g., Starcoder mixed in with 20%).
3. Related, is it possible to distill the findings from Fig. 4 into a simple heuristic that relates the mixture weight to the midtraining start (possibly as a fraction of total training) and possibly midtraining data size? Such a heuristic might be a helpful rule of thumb for developers.
4. I find the observation in Sec. 4 that in-domain improvements and C4 retention are aligned (looking at Tab. 2, we generally even have improvements in C4 validation loss) interesting, maybe even somewhat counterintuitive and believe they merit a bit more attention. Do the authors have any ideas where this is coming from? I would usually assume C4 validation loss to increase when any other data source is included in training.
5. Fig. 4 only considers SFT loss, what about C4 val loss, do we see the same interaction? Fig. 4 would suggest that high weight early is best, but is that also best for overall performance? Presumably C4 validation would suffer here?
6. How do you expect results would translate to multiple midtraining stages with multiple data sources, possibly introduced at different times?

Most of these points are merely clarifying questions / curious considerations. I believe the paper is in a very good state already.

# Minor Issues / Comments
- missing verb in L 50 (right)
- inconsistent punctuation in paragraph headings
- greying out data points in Fig. 3 is not effective when one of the salient SFT Dataset colors is also grey (PyCode), please change for clarity (or is "greyed out" here meant figuratively to indicate some data points were omitted?).

---

> ### Author Rebuttal · Authors · 2026-03-31
>
> We thank you for appreciating our work and for the thorough reading and engagement! We are happy that you found the theoretical framing clean and the presentation to be strong. We address your clarifying questions below.
>
> **W1:  On the proximity term formulation**
>
> We will add an ablation in the appendix reporting the proximity advantage vs. improvement result for each individual component (cosine similarity, vocabulary Jaccard, JS-divergence). We also clarify that the weighted combination emerged after experimentation: embedding-based similarity produced implausible rankings for code-heavy datasets [imgur link], and gradient-based methods were unstable across layers and examples. We expect the combined metric to be robust to reasonable alternative formulations, and use the unigram based metric as more of a minimal working example.
>
> **W2: On the default mixture weight**
>
> These were driven by data availability mainly, as we wanted to remove the variable of repetition as repeating the same data many times is known to be harmful to LM performance in pretraining [cite]. As seen in Table 1, different domains had very different numbers of available tokens, and we chose the default mixture weights accordingly. That said, we did also try the timing x mixture ablation in which we explicitly varied mixture weights.
>
> **W3: On a practical heuristic**
>
> Based on our findings, a qualitative rule of thumb is: earlier introduction of specialized data supports more aggressive mixture weights, while late introduction favors conservative mixing. Tentatively, we conjecture that domains more distant from the general pretraining distribution would benefit more from early introduction while it may not matter for more similar domains. This is what our theoretical framing would suggest, and we plan to investigate this empirically in future work.
>
> **W4/W5: In-domain vs. C4 loss**
>
> It’s true that there’s usually a tradeoff between in-domain performance and retention of old tasks, but in this setting (midtraining -> posttraining) these two things are pretty well correlated. This is the correlation per dataset:
>
> | Dataset| Pearson (r) | Two-sided (p)-value |
> |---|---:|---:|
> | GSM8k| 0.9296 | 5.22×10^-11 |
> | LIMA | 0.9936 | 2.34×10^-22 |
> | Pycode| 0.7357 | 4.19×10^-5 |
> | SciQ | 0.9111 | 6.20×10^-10 |
>
> The trends for C4 loss on the timing x weight ablation are also accordingly pretty similar, we will add these results to the appendix.
>
> **70M (PyCode)**
> | Start (steps) | Mix Weight | SFT Val Loss | C4 Val Loss |
> |---|---|---|---|
> | 13B | 80% | 2.3783 | 6.0676 |
>  | 13B | 20% | 2.5036 | 6.0318 |
>  | 13B | 10% | 2.5144 | 5.9856 |
> | 42B | 80% | 2.4373 | 6.0287 |
> | 42B | 10% | 2.5518 | 6.0772 |
> | 63B | 80% | 2.5673 | 6.1653 |
> | 63B | 30% | 2.5080 | 5.9744 |
>  | 63B | 20% | 2.6235 | 6.1321 |
> | 63B | 10% | 2.5804 | 6.0612 |
>  | 40k | 30% | 2.6293 | 6.1078 |
> | 40k | 10% | 2.6114 | 6.0980 |
>  | 105B | 80% | 2.7603 | 6.3293 |
>  | 105B | 10% | 2.6762 | 6.1622 |
>
> **160M (PyCode)**
>  | Start (steps) | Mix Weight | SFT Val Loss | C4 Val Loss |
>  |---|---|---|---|
>  | 13B | 80% | 1.9772 | 5.0881 |
> | 13B | 20% | 2.1340 | 5.0785 |
> | 13B | 10% | 2.1751 | 5.1295 |
>  | 42B | 80% | 2.0477 | 5.1206 |
>  | 42B | 10% | 2.2671 | 5.3196 |
>  | 63B | 80% | 2.1457 | 5.1383 |
> | 63B | 30% | 2.2053 | 5.1402 |
> | 63B | 20% | 2.2192 | 5.1517 |
> | 63B | 10% | 2.2619 | 5.1990 |
> | 40k | 30% | 2.3952 | 5.4232 |
> | 40k | 10% | 2.2673 | 5.1644 |
> | 105B | 80% | 2.4611 | 5.8716 |
> | 105B | 10% | 2.3174 | 5.2554 |
>
> We believe that this stems from the bridging framing, as when midtraining reduces the “effort” to reach the SFT optimum, less representational disruption also happens overall, which preserves performance.
>
> **W6: On multiple midtraining stages**
>
> This is an interesting direction we leave to future work. Our framework predicts that sequential stages would be beneficial if each intermediate distribution progressively bridges toward posttraining, but testing this is beyond our current scope.

---

> > ### Author Rebuttal · Reviewer_gFQ8 · 2026-04-06
> >
> > I thank the authors for their insightful comments. While the links seem to not be included correctly (the [imgur link] and [cite]), my concerns are adequately addressed. Given that I already strongly recommend acceptance, I am maintaining my score.

---

> > > ### Author Response · Authors · 2026-04-07
> > >
> > > Thanks for bringing the links to our attention. These were meant to be added when drafting the response but we forgot to do so. This is purely informational, but just in case you are interested:
> > >
> > > [imgur link] - should be https://imgur.com/a/Njtwn18 . We found the early embedding-based similarity (from Qwen3-8B embedding) significantly less aligned with intuition (e.g. there is a low similarity between Starcoder and Pycode), while the gradient based alignment (Pythia-1b shown) had similar issues and varied by layer substantially.
> > >
> > > [cite] - this should be https://arxiv.org/pdf/2205.10487 I believe

---

### Official Review · Reviewer_eYcc · 2026-03-12

**Soundness:** 3
**Presentation:** 4
**Significance:** 3
**Originality:** 4
**Overall Recommendation:** 5
**Confidence:** 3

**Summary:**

This paper is primarily an empirical study of midtraining. Its central claim is that midtraining helps post-training mainly by providing a better initialization for the subsequent post-training stage. To support this view, the paper presents a short theoretical derivation together with a systematic experimental study.

**Compliance With Llm Reviewing Policy:**

Affirmed.

**Final Justification:**

I do not have any fundamental concerns about the paper; my main comments were primarily related to the writing and presentation. Since the authors have committed to revising these aspects, I do not have further concerns at this point. I therefore maintain my recommendation in favor of acceptance.

**Key Questions For Authors:**

N/A

**Strengths And Weaknesses:**

### Strength:
- The paper studies an important and timely problem. Midtraining is widely used in practice, yet its role is still poorly understood. A systematic study of when it helps, what kinds of data are useful, and how it should be scheduled is therefore valuable.
- The empirical study is thorough and well organized. The paper investigates several aspects of midtraining in a controlled way, including domain specificity, distributional proximity, comparison with continued pretraining, and the interaction between timing and mixture weight.
- The paper provides several concrete and practically relevant takeaways. It shows that midtraining is especially helpful in high-shift domains such as code and math, that it can outperform naive continued pretraining, and that scheduling choices can matter as much as, or more than, mixture weight. These findings are likely to be useful for future practice.

### Weaknesses:
The main mechanism proposed in this paper, i.e. that midtraining helps post-training by finding a better initialization for it, seems vague and not fully convincing to me. In particular, I think the authors need to address the following issues:
- First, " providing better initialization" is a very vague statement. Any helpful operation on the parameters can be said to provide better initialization for the next stage of optimization. In this sense, pretraining itself can also be said to help post-training by providing a better initialization. As a key claim, this feels too generic and does not provide much real insight into what is special about midtraining.
- Second, the derivation in Section 2.3 does not fully establish this point. The authors seem to argue that midtraining, by using data closer to post-training, reduces the term $J_T(\theta_0)$ in the effort term. However, since the trajectory $\theta_t$ also depends on $\theta_0$, changing $\theta_0$ can also affect the alignment term. I do not see any evidence showing which of these two effects is more important.
- Third, this claim does not clearly distinguish midtraining from post-training itself. If one directly uses all post-training data as midtraining data without mixing in pretraining data (essentially just make midtraining same as post-training but under a different name), then much of the paper's central narrative would still seem to apply. This suggests that the proposed explanation does not yet identify what is truly distinctive about midtraining as a separate stage.

My understanding is that the authors aim to build a systematic study around the question of why midtraining works, and to provide an answer to that question. In my view, the exploration itself is clearly valuable, because it systematically verifies many important properties of midtraining and can inspire future practice. However, the specific answer offered by the paper is, for the reasons above, not fully satisfactory. I would suggest reorganizing the paper more as a practically oriented study of midtraining, rather than insisting on giving a definitive explanation of why midtraining works.

---

> ### Author Rebuttal · Authors · 2026-03-31
>
> We thank you for your thoughtful comments on our paper and for your appreciation of the practical takeaways. We address your conceptual concerns below:
>
> **W1: On better initialization being too generic**
>
> We agree that the verbal description of better initialization is generic, but the description in the paper provides a more specific interpretation: midtraining improves initialization by moving $\theta_0$ closer to the SFT optimum along the distributional gradient formalized through the effort term $J_T(\theta_0) - J^*_T$ in our bound while preserving general capabilities through the pretraining mixture. This is distinct from pretraining, which also "provides better initialization" for posttraining but does not specifically bridge toward the target distribution. The proximity advantage metric operationalizes exactly this specificity: not all parameter changes reduce the effort term, only those that move the model toward the target distribution while maintaining the general data mixture. We will clarify this distinction in the revision.
>
> **W2: On distinguishing midtraining from posttraining**
>
> This is an interesting point, but we believe our working definition of midtraining in Section 2.1 already addresses it. We define midtraining as an intermediate phase that (1) occurs between pretraining and posttraining, and (2) maintains a mixture with general pretraining data, meaning that using all post-training data without mixing in general data would, under our definition, simply be posttraining itself, not midtraining. The three-stage structure (pretraining ->  midtraining -> posttraining) is what makes midtraining a meaningful concept distinct from the other phases. This is also why we evaluate after posttraining, rather than directly after pretraining/midtraining.
>
> **W3: On reframing as a practical study**
>
> We are sympathetic to this suggestion and will soften the framing of the central claim to accurately reflect the scope of the theory, that is, as a formal sketch consistent with our findings rather than a definitive explanation. However, we will still retain the theoretical contribution as a motivating framework that generates testable predictions. We believe this framing is faithful to what the paper actually delivers, and that the extended analysis strengthens the case for keeping the theory in the paper.

---

> > ### Author Rebuttal · Reviewer_eYcc · 2026-04-04
> >
> > Thank you for the authors' response. I do not have any fundamental concerns about the paper; my main comments were primarily related to the writing and presentation. Since the authors have committed to revising these aspects, I do not have further concerns at this point. I therefore maintain my recommendation in favor of acceptance.

---

### Official Review · Reviewer_E72R · 2026-03-13

**Soundness:** 2
**Presentation:** 2
**Significance:** 3
**Originality:** 3
**Overall Recommendation:** 4
**Confidence:** 3

**Summary:**

The authors of this paper conduct a systematic study of midtraining to identify the source of its benefits relative to other techniques, such as continual pretraining, etc. They derive a mathematical bound on catastrophic forgetting and show that it can be reduced by better initialization, which is achieved by midtraining. The authors then conduct extensive experiments to evaluate the effects of midtraining data domain compatibility, midtraining start timing, and mixture weights on final performance. They also show that midtraining outperforms continual pretraining even with the smaller number of domain-specific tokens ever seen during training.

**Compliance With Llm Reviewing Policy:**

Affirmed.

**Final Justification:**

At the beginning, the paper felt a little disconnected. Theoretical analysis wasn't fully grounded in empirical observations, there were considerable differences in the results for different model sizes (which are quite small, the biggest is only 1 billion parameters), and the overall effect of midtraining was not easily distinguishable from other possible factors.

Given the author's most recent response, I feel significantly more convinced in the paper's results and ability to explain what benefits midtraining brings to the training dynamics. There are still some concerns about the notation, applicability to larger models, etc. Therefore, my final score is 4 (Weak Accept), assuming the authors will do significant job to polish their work and integrate new results into it.

**Key Questions For Authors:**

**Q1.** While I understand the practical importance of mitigating catastrophic forgetting, what is the purpose of measuring and optimizing the $\Delta_P(K)$ from the theoretical standpoint? Isn’t the final objective meant to achieve the optimal loss on the post-training population - $J^{\*}_T$, which is represented by $\theta^{\*}_T$? Then, if $\theta^{\*}_T$ is the optimal model for the post-training population, the final performance on the pretraining distribution is fixed at $J_P(\theta^{\*}_T)$. This question is out of genuine curiosity, given the considerable time you could have devoted to the theoretical analysis and the potential insights it might have yielded.

**Q2.** What do derived bounds in Section 2.3 and Appendix A contribute to this work?

**Q3.** Do you find the language modeling loss on the benchmarks a reliable metric of the model’s performance?

**Q4.** Could you elaborate on your findings in Section 7? I don’t see any noticeable difference between “Base vs Base-SFT” and “Midtrained vs Midstrained-SFT” in Figure 6. Is the last layer the most important in the CKA analysis you conducted? Other layers do not seem to have a meaningful trend.

**Q5.** What is “Num ranks” in Table 4?

**Q6.** Do experiments in Section 5.2 intentionally use a different number of domain-specific tokens between midtraining and continual pretraining? Continual pretraining uses 128B _(the total training run)_ - 83B _(starting point of continual pretraining)_ = 45B tokens from Starcoder. At the same time midtraining uses only (128B - 12.6B _(starting point of midtraining)_) * 0.2 _(mixture weight)_ = 23.08B tokens.

**Limitations:**

The limitations are discussed in the last sentence of the conclusions, but would benefit from being moved to a separate section.

**Strengths And Weaknesses:**

**Strengths:**

**S1.** A substantial number of large-scale computationally heavy pretraining experiments.

**S2.** Important and timely empirical findings showing that midtraining on fewer domain-specific tokens outperforms continual pretraining with almost twice as many domain-specific tokens.

**Weaknesses:**

**W1.** The paper provides a theoretical analysis that claims to explain how midtraining can “mitigate forgetting on the pretraining distribution”. It concludes that the initialization $\theta_0$ can change the upper bound on forgetting $\Delta_P(K)$ by being closer to $J^*_T$. However, $\theta_0$ not only affects the bound, but the $\Delta_P(K)$ as well, which is defined as $\Delta_P(K) := J_P(\theta_K) - J_P(\theta_0)$. This is done throughout Appendix A and Section 2.3. This way, $\Delta_P(K)$ is the measure of forgetting during post-training, but it cannot be directly compared, since a lower bound for post-training after midtraining does not account for the forgetting during the midtraining stage.

**W2.** The theoretical analysis introduces the concepts of plasticity and other lemmas in Appendix A, and derives an updated bound in Equation 19. This, however, is never used or even mentioned in the experiments or subsequent sections, thereby disassociating it from the rest of this paper. Appendix A by itself does not come to any conclusions. Section 2.3 restates that the only influence the midtraining has is the initialization $\theta_0$. It should either be removed (which I would like to avoid) or be meaningfully completed and built upon.

**W3.** I am not convinced that measuring the language modeling loss on the benchmarks is a good proxy of the model’s performance on them. I can see how one model can outperform the other one by being more aligned in terms of the answer’s format and style (e.g., CoT-style answers for GSM8K), but achieve significantly lower accuracy. While I understand that the tested model sizes might lack the capability to be meaningfully evaluated on benchmarks like GSM8K, it would provide a more trustworthy perspective on the findings presented in this paper.

**W4.** While I appreciate the effort and resources dedicated to conducting the experiments presented in the paper, it seems the authors should at least opt to use a 1-billion-parameter model in most of them to improve stability (which has already been experimented with). This could help alleviate **W3**, and other stability issues: for example, in Figure 4, the conclusions may be quite different when comparing 70m and 160m models. At 60B tokens, the ranking of mixture weight is rather arbitrary (30% performing best, 20% worst, 80%, and 10% in the middle). This already changes into more intuitive results in the case of the 160m model. Using a bigger model would remove any doubts about the trustworthiness of the reported findings.

**W5.** General presentation issues:

- **W5.1.** Line 49 (right column): “Our results that midtraining appears …” - add “show”.
- **W5.2.** Line 250 (left column) reports the correlations “r = 0.869”, which is “particularly strong for smaller models”, whereas Figure 3 reports “r = 0.521”, which is smaller than two out of three bigger models (160m and 1b).
- **W5.3.** Lines 336-339 (left column) present the training times in steps, while discussing Figure 5, whose x-axis is in tokens. This would benefit from unification.
- **W5.4.** Figure 5 does not have a legend.
- **W5.5.** Line 826 mentions “All three models were trained …”, discussing Table 4, which lists 4 models.

If my concerns are properly addressed, I am ready to raise my score.

---

> ### Author Rebuttal · Authors · 2026-03-31
>
> We thank you for the detailed reading. We believe the concerns raised are addressable and address each below.
>
> **W1/W2/Q1/Q2**
>
> We first clarify scope: Section 2.3 explicitly frames the analysis as "a simple theoretical sketch that formalizes the core intuition," and Appendix A notes we "do not claim that these assumptions hold globally." The theory is meant to show the distributional bridging interpretation is formally coherent, not to provide a complete mechanistic account. We will state this more clearly.
>
> **W1:**
>
> You raise a precise point: since $\Delta_P(K) := J_P(\theta_K) - J_P(\theta_0)$ and midtraining changes $\theta_0$, a lower $\Delta_P(K)$ after midtraining could simply reflect an elevated $J_P(\theta_0)$ baseline, i.e., midtraining itself caused forgetting, rather than less forgetting during posttraining. The bound therefore cannot be directly compared across conditions.
>
> We agree this is subtle and clarify on two fronts. First, the bound is explicitly scoped to the posttraining phase: given a fixed $\theta_0$, it characterizes how much forgetting posttraining induces. The claim is that successful midtraining provides a $\theta_0$ from which posttraining causes less forgetting. The midtraining stage's own effect on $J_P$ is separate, and we do not claim the bound accounts for it. We will make this explicit in the revision.
>
> Empirically, Table 2 addresses this concern: C4 validation loss after midtraining is largely preserved, so midtraining does not substantially elevate $J_P(\theta_0)$. Therefore, comparing $\Delta_P(K)$ across conditions is meaningful in our setting. Taken together, the theory characterizes posttraining forgetting given $\theta_0$, and the empirics confirm that $\theta_0$ itself is not materially degraded by midtraining.
>
> **W2**
>
> Equation 19 relates to Finding 4: it bounds forgetting as a function of start time $t$ through $S(t) = \sum_u s(t+u)$, where $s(\cdot)$ is nonincreasing, and mixture weight $w$ via $G(w)$. Later start times reduce $S(t)$, limiting how much initialization can improve; larger $w$ increases $G(w)$, amplifying the cost through the $G(w)^2 S(t)^2$ term. The specific interaction we observe empirically that late introduction with high mixture weight is uniquely harmful follows from this multiplicative structure. This is also consistent with (https://arxiv.org/abs/2503.19206), overtraining/loss of plasticity, which we interpret as a manifestation of reduced $S(t)$ at late start times. We will edit the writing to make this connection more explicit.
>
> **Q1:** Thank you for this question. Why is bounding $\Delta_P(K)$ meaningful if $\theta_T^{\ast}$ is the posttraining optimum, where pretraining performance is already fixed? In practice, we never reach $\theta_T^{\ast}$. Forgetting is therefore a finite-horizon problem, and the bound characterizes how initialization affects that trajectory.
>
> **Q2:** We address the alignment term in the context of W1 above: both terms benefit from the same initialization, since a smaller effort term implies the trajectory stays close to $\theta_0$, preserving any gradient alignment that midtraining has established.
>
> **W3:** We thank the reviewer for this important point. We agree that accuracy is a more direct measure of downstream utility, and we provide SciQ accuracy results here to address this concern.
>
> For the SciQ-finetuned models, midtrained models (knowledgeQA)  show consistent accuracy improvements over baselines: +1.78pp (70M), +0.92pp (160M), and +1.08pp (410M), averaged across 5 seeds. At 1B, both models are around the same, but we note there is also not much difference for SciQ’s loss in the 1B model (Table 2).
>
> For GSM8K specifically, we agree with the reviewer that accuracy is unreliable at sub-1B scale. GSM8K accuracy is near chance (~2-4%) for both conditions, with overlapping confidence intervals. For small models, cross-entropy loss on held-out text is usually used as a more reliable signal for generative tasks. We know from past work that midtraining does work at larger scales, so we will augment our findings with larger models in future.
>
> **W4:** Use of larger models in ablations. We note that the directional trends in Figure 4 are already consistent across 70M and 160M. We agree larger-scale ablations would strengthen confidence and will expand the timing-weight sweep to 410M/1B in the camera-ready.
>
> **Presentation issues.** Thank you for your careful reading, we will fix all issues. Several of the issues were leftover text from an older version of the paper, we will double check the other results to make sure they match the current version.
>
> **Q5** "Num ranks" refers to the number of GPUs per node in our training setup.
>
> **Q6** The token count asymmetry is intentional and is the point of Section 5.2: midtraining outperforms continued pretraining even with substantially fewer domain-specific tokens. We will make this design choice more prominent in the text.

---

> > ### Author Rebuttal · Reviewer_E72R · 2026-04-02
> >
> > Thank you for your extensive response.
> >
> > Some of my concerns were partially addressed: “a simple theoretical sketch that formalizes the core intuition” indeed slightly softens the critique towards the shift in the meaning of $\theta_0$, and the perspective of analyzing only the post-training and how it affects the forgetting can be understandable, but it needs to be explicitly noted (potentially by using a separate variable), otherwise the current version is highly misleading.
> >
> > At the same time, I find the **Q1** by reviewer **MHUz** very relevant, and one of the few things that may empirically support the theoretical analysis. Since Equation 5 (12) introduces two bound components: the gradient alignment term and the effort term, the results from Table 2 you cited seem to explain the effort term, assuming that the initialization point after the relevant midtraining mixture has better SFT loss than after other ones or no midtraining. Therefore, it would be beneficial if you could also report the losses before the SFT training, so we could evaluate the forgetting introduced by the post-training itself, as you frame it in your theoretical analysis. Furthermore, adding gradient alignment comparison would complete the empirical evaluation of the derived bound.
> >
> > Other than that, I appreciate that the 5000-character limit is quite restrictive, but could you also address **Q4**?
> >
> > **upd.** Given the author's most recent response, I feel significantly more convinced in the paper's results and ability to explain what benefits midtraining brings to the training dynamics. There are still some concerns about the notation, applicability to larger models, etc. Therefore, I am raising my score from 2 to 4 to reflect this.

---

> > > ### Author Response · Authors · 2026-04-06
> > >
> > > Thanks for following up, we agree that the current notation may make the theory unclear, and we will revise the draft so that the bound is stated to be about posttraining forgetting conditioned on the initialization $\theta_0$, rather than total forgetting across both midtraining and posttraining.
> > >
> > > **On the effort term/pre-SFT losses**
> > >
> > > Following your suggestion, we computed losses before SFT for the main beneficial midtraining conditions. These show that matched midtraining improves the target domain initialization (for code/math) while leaving held out C4 almost unchanged.
> > >
> > > Code case:
> > >
> > > | Model | Base PyCode | Code-mid PyCode | Δ(PyCode) | Base C4 | Code-mid C4 |
> > > |---|---:|---:|---:|---:|---:|
> > > | 70M  | 5.441 | 3.248 | -2.193 | 4.121 | 4.160 |
> > > | 160M | 4.798 | 2.561 | -2.237 | 3.634 | 3.664 |
> > > | 410M | 4.459 | 2.136 | -2.323 | 3.256 | 3.254 |
> > > | 1B   | 4.151 | 1.948 | -2.203 | 3.002 | 3.032 |
> > >
> > > Math case:
> > > | Model | Base GSM8K | Math-mid GSM8K | Δ(GSM8K) | Base C4 | Math-mid C4 |
> > > |---|---:|---:|---:|---:|---:|
> > > | 70M  | 4.100 | 2.653 | -1.447 | 4.121 | 4.162 |
> > > | 160M | 3.590 | 1.918 | -1.672 | 3.634 | 3.646 |
> > > | 410M | 3.163 | 1.429 | -1.734 | 3.256 | 3.265 |
> > > | 1B   | 2.886 | 1.301 | -1.585 | 3.002 | 2.998 |
> > >
> > > This supports the intended interpretation of the "effort term" wherein domain matched midtraining gives a better posttraining initialization for that SFT domain rather than lowering the starting C4 loss substantially.
> > >
> > > **Posttraining-only forgetting**
> > >
> > > Using the pre-SFT measurements above, we can now compute the posttraining-only forgetting quantity, which the theoretical sketch describes. E.g. for the 1b model,
> > >
> > > | Setting | Pre-SFT C4 | Post-SFT C4 | Posttraining forgetting Δ |
> > > |---|---:|---:|---:|
> > > | 1B base → PyCode SFT | 3.002 | 3.075 | 0.073 |
> > > | **1B code-midtrained → PyCode SFT** | **3.032** | **3.070** | **0.038** |
> > > | 1B base → GSM8K SFT | 3.002 | 3.134 | 0.132 |
> > > | **1B math-midtrained → GSM8K SFT** | **2.998** | **3.018** | **0.020** |
> > >
> > > So in our setting, matched midtraining improves the target initialization as well as forgetting incurred during SFT.
> > >
> > >
> > > **Alignment term/gradient conflict**
> > >
> > > We also ran an empirical alignment analysis. Direct gradient-cosine estimates were somewhat noisy at these scales, so we used a finite-difference probe where we take a small step in the target-SFT gradient direction and measure the induced change in held out C4 loss. This is a measurement of the one-step forgetting effect on the left hand side of our bound, i.e. $J_P(\theta_{t+1}) - J_P({\theta_t})$. We want mid ΔJ_C4 to be lower than base ΔJ_C4 in the table below, and a negative number means that moving in the target update direction of the SFT dataset would lower C4 loss as well.
> > >
> > > | Model | base ΔJ_C4 | mid ΔJ_C4 | ratio (mid/base) |
> > > |---|---:|---:|---:|
> > > | 70m_code  | +3.1e-4 | -2.2e-5 | -0.07× |
> > > | 70m_math  | +1.2e-3 | +2.4e-3 | +1.89× |
> > > | 160m_code | +6.0e-5 | -4.3e-5 | -0.71× |
> > > | 160m_math | +2.0e-5 | -5.3e-5 | -2.68× |
> > > | 410m_code | +1.2e-4 | +2.1e-5 | +0.18× |
> > > | 410m_math | +6.7e-5 | +4.1e-6 | +0.06× |
> > > | 1b_math   | +7.2e-6 | -1.6e-5 | -2.26× |
> > > | 1b_code   | +8.6e-5 | +3.0e-5 | +0.35× |
> > >
> > > In the main successful regimes, this target step causes substantially less immediate C4 degradation from the midtrained initialization than from the base initialization. This is qualitatively consistent with the alignment/conflict term in the theory: in the successful midtraining settings, the target update direction is less harmful to the pretraining objective at initialization. The notable exception is 70M-math, which is unsurprising as the C4 loss for 70M-math is actually not lower in the main results (see Table 8).
> > >
> > > **Q4/CKA analysis**
> > >
> > > the intended claim is not that every layer shows a large separation, but that the most meaningful differences are concentrated in later layers. This was based on the finding (https://arxiv.org/abs/2202.10054) that shows that one signature of feature distortion during finetuning and subsequent forgetting of pretraining data is heavy changes to the final layer. We will revise the text to make that takeaway clearer and avoid overgeneralizing beyond what Figure 6 shows.

---

### Decision · Program_Chairs · 2026-04-30

**Decision:**

Accept (spotlight)

**Comment:**

The reviewers agreed that the paper tries to understand a very relevant and important component of the LLM training pipeline, and contributes critical practical advise on data mixing ratios, and introduction of specialized data. They all noted the importance and novelty of the “finding 4”: the observation that early introduction of specialized data supports aggressive mixture weights; in contrast, late introduction high mixture weights are harmful. All reviewers complimented the experimental design and ablations. The presentation is clear and well structured.